

# Production of N₂O₅ and ClNO₂ in summer in urban Beijing, China

Wei Zhou[1,2#], Jian Zhao[1,2#], Bin Ouyang[3], Archit Mehra[4], Weiqi Xu[1,2], Yuying Wang[5], Thomas J. Bannan[4], Stephen D. Worrall[4,a], Michael Priestley[4], Asan Bacak[4], Qi Chen[6], Conghui Xie[1,2], Qingqing Wang[1], Junfeng Wang[7], Wei Du[1,2], Yingjie Zhang[1], Xinlei Ge[7], Penglin Ye[8,11], James D. Lee[9], Pingqing Fu[1,2], Zifa Wang[1,2], Douglas Worsnop[8], Roderic Jones[3], Carl J. Percival[4,b], Hugh Coe[4], Yele Sun[1,2,10]

[1]State Key Laboratory of Atmospheric Boundary Layer Physics and Atmospheric Chemistry, Institute of Atmospheric Physics, Chinese Academy of Sciences, Beijing 100029, China

[2]University of Chinese Academy of Sciences, Beijing 100049, China

[3]Department of Chemistry, University of Cambridge, Cambridge CB2 1EW, UK

[4]Centre for Atmospheric Science, School of Earth, Atmospheric and Environmental Science, University of Manchester, Manchester M13 9PL, UK

[5]College of Global Change and Earth System Science, Beijing Normal University, Beijing 100875, China

[6]College of Environmental Sciences and Engineering, Peking University, Beijing 100871, China

[7]School of Environmental Science and Engineering, Nanjing University of Information Science and Technology, Nanjing 210044, China

[8]Aerodyne Research, Inc., Billerica, Massachusetts 01821, USA

[9]National Centre for Atmospheric Science, University of York, Heslington, York YO10 5DD, UK

[10]Center for Excellence in Regional Atmospheric Environment, Institute of Urban Environment, Chinese Academy of Sciences, Xiamen 361021, China

[11]Nanjing DiLu Scientific Instrument Inc, Nanjing, 210036, China.

[a]Now at School of Materials, University of Manchester, M13 9PL, UK

[b]Now at Jet Propulsion Laboratory, 4800 Oak Grove Drive, Pasadena, CA 91109

[#]These authors contributed equally to this work

***Correspondence***: Yele Sun (sunyele@mail.iap.ac.cn) and Hugh Coe (hugh.coe@manchester.ac.uk)



**Abstract.** The heterogeneous hydrolysis of dinitrogen pentoxide ($N_2O_5$) has a significant impact on both nocturnal

particulate nitrate formation and photochemistry the following day through photolysis of nitryl chloride ($ClNO_2$), yet these

processes in highly polluted urban areas remain poorly understood. Here we present measurements of gas-phase $N_2O_5$ and

$ClNO_2$ by high-resolution time-of-flight chemical ionization mass spectrometers (ToF-CIMS) during summer in urban

Beijing, China as part of the Air Pollution and Human Health (APHH) campaign. $N_2O_5$ and $ClNO_2$ show large day-to-day

variations with average ($\pm 1\sigma$) mixing ratios of 79.2 ± 157.1 and 174.3 ± 262.0 pptv, respectively. High reactivity of $N_2O_5$,

with $\tau(N_2O_5)^{-1}$ ranging from $0.20 \times 10^{-2}$ to $1.46 \times 10^{-2}$ $s^{-1}$, suggests active nocturnal chemistry and a large nocturnal nitrate

formation potential via $N_2O_5$ heterogeneous uptake. The life time of $N_2O_5$, $\tau(N_2O_5)$, decreases rapidly as the increase of

aerosol surface area, yet it varies differently as a function of relative humidity with the highest value peaking at ~40%.

The $N_2O_5$ uptake coefficients estimated from the product formation rates of $ClNO_2$ and particulate nitrate are in the range of

0.017-0.19, corresponding to direct $N_2O_5$ loss rates of 0.00044-0.0034 $s^{-1}$. Further analysis indicates that the fast $N_2O_5$ loss in

the nocturnal boundary layer in urban Beijing is mainly attributed to its indirect loss via $NO_3$, for example through the

reactions with volatile organic compounds and NO, while the contribution of heterogeneous uptake of $N_2O_5$ is comparably

small (7-33%). High $ClNO_2$ yields ranging from 0.10 to 0.35 were also observed which might have important implications

for air quality by affecting nitrate and ozone formation.



# 1    Introduction

Dinitrogen pentoxide ($N_2O_5$) is an efficient sink for the nocturnal removal of nitrogen oxides ($NO_x$) (Dentener and Crutzen, 1993; Brown et al., 2006), and exists in a rapid temperature-dependent thermal equilibrium with nitrate radical ($NO_3$) – one of the most important oxidants at night-time (Wayne et al., 1991). Although $NO_3$ and $N_2O_5$ levels can be suppressed by rapid

titration of $NO_3$ against NO and volatile organic compounds (VOCs) in urban areas (Brown et al., 2003b), heterogeneous uptake by aerosol particles, fog and cloud droplets is often found to be the major pathway for direct $N_2O_5$ removal (Thornton et al., 2003; Brown et al., 2006; Bertram and Thornton, 2009; Chang et al., 2011; Wagner et al., 2013). $N_2O_5$ can produce nitryl chloride ($ClNO_2$) on chloride–containing aerosols which serves as an important reservoir of $NO_x$ (Finlayson-Pitts et al., 1989; Thornton et al., 2010; Phillips et al., 2012). It has been found that levels of $HNO_3$ formed through hydrolysis of $N_2O_5$

at night-time were comparable to the levels produced from the reaction of $NO_2$ with OH radical during daytime (Geyer et al., 2001). Furthermore $N_2O_5$ and $ClNO_2$ can be photolyzed into $NO_2$ and atomic chlorine (Cl) after sunrise, resulting in significant impacts on daytime photochemistry, for example trace gas degradation and ozone formation (Osthoff et al., 2008; Riedel et al., 2012; Mielke et al., 2013; Sarwar et al., 2014). Thus, it is of great importance to understand $N_2O_5$ and $ClNO_2$ chemistry in the nocturnal boundary layer of various environments.

The heterogeneous reaction of $N_2O_5$ and activation of $ClNO_2$ are parameterized by the $N_2O_5$ uptake coefficient ($\gamma_{N_2O_5}$) and $ClNO_2$ yield (ø), which are defined as the reaction probability of $N_2O_5$ upon its collision on an aerosol surface and the number of $ClNO_2$ molecules formed per lost $N_2O_5$ molecule upon uptake, respectively (Brown et al., 2006; Roberts et al., 2009; Wagner et al., 2013). Previous laboratory studies have shown a large variability of $\gamma_{N_2O_5}$ (0.0002-0.3) depending on the physical characteristics of the substrates (e.g., aerosol surfaces, water droplets, and ice/crystal surfaces), environmental

conditions (e.g., acidity, relative humidity and temperature), and chemical composition of aerosol particles (e.g., nitrate, sulfate, black carbon and organic coating) (Thornton and Abbatt, 2005; Anttila et al., 2006; McNeill et al., 2006; Sander et al., 2006; Cosman et al., 2008; Chang et al., 2011). To reveal the effects of each factor on $N_2O_5$/$ClNO_2$ chemistry, several parameterizations of $\gamma_{N_2O_5}$ and ø have been proposed during the last decade (Riemer et al., 2003; Evans and Jacob, 2005; Anttila et al., 2006; Davis et al., 2008; Griffiths et al., 2009; Riemer et al., 2009). For example, Bertram and Thornton (2009)




constructed a parameterization of $\gamma_{N_2O_5}$ as a function of aerosol liquid water, nitrate, and chloride content based on the measurements of laboratory-generated internally mixed chloride-nitrate particles. Similarly, ø was parameterized as a function of aerosol liquid water content and aerosol chloride (Roberts et al., 2009). These results have great implications for regional/global chemical transport models which aim to improve the simulations of nitrate and ozone (Evans and Jacob,

2005; Sarwar et al., 2014). However, the field-derived values of $\gamma_{N_2O_5}$ and ø often exhibit large inconsistencies with laboratory results, suggesting a more complex nature of heterogeneous $N_2O_5$ uptake in the ambient atmosphere (Chang et al., 2011).

      $N_2O_5$ and $NO_3$ can be measured by various different techniques which have been summarized in Chang et al. (2011). For example, $N_2O_5$ can be derived from the thermal equilibrium with $NO_2$ and $NO_3$ that are simultaneously measured by

differential optical absorption spectroscopy (DOAS) (Platt and Stutz, 2008; Stutz et al., 2004). Another indirect measurement of $N_2O_5$ is subtracting ambient $NO_3$ from the total measured $NO_3$ after converting $N_2O_5$ to $NO_3$ in a heated inlet (O'Keefe and Deacon, 1988; Smith et al., 1995; Brown et al., 2001; Wood et al., 2003; Stutz et al., 2010). It should be noted that $N_2O_5$ measured with these two approaches may introduce additional uncertainties because it is not directly measured. $N_2O_5$ can be directly measured by chemical ionization mass spectrometer (CIMS) using iodide ($I^-$) as the reagent ion (I−CIMS) with high

sensitivity and time resolution (Slusher et al., 2004; Zheng et al., 2008; Kercher et al., 2009). However, simultaneous measurements of $N_2O_5$ and $NO_3$ using thermal dissociation (TD)–CIMS need to consider the interference of $m/z$ 62 ($NO_3^-$) from thermal decomposition of peroxy acetyl nitrate (PAN) and other related species (Wang et al., 2014). The I-CIMS is also widely used to measure $ClNO_2$ in both laboratory and field studies (Thornton and Abbatt, 2005; McNeill et al., 2006; Osthoff et al., 2008). A large amount of $ClNO_2$ was first observed in polluted coastal regions owing to the abundant chloride from sea

salt aerosol, for example, the Gulf of Mexico and the Los Angeles basin (Osthoff et al., 2008; Kercher et al., 2009; Riedel et al., 2012). High levels of $ClNO_2$ from anthropogenic chloride sources were also reported in some inland areas (Thornton et al., 2010; Mielke et al., 2011; Phillips et al., 2012; Bannan et al., 2015; Phillips et al., 2016). More recently, several studies in Hong Kong (Tham et al., 2014; Brown et al., 2016b; Wang et al., 2016a) and in the North China Plain (NCP) (Tham et al., 2016; Wang et al., 2017b; Wang et al., 2017c; Wang et al., 2018) observed consistently high mixing ratios of $N_2O_5$ and

$ClNO_2$. In particular, $ClNO_2$ can be rapidly formed in the plumes of coal-fired power plants in the NCP, which serves as an



important source of chloride in non-ocean regions. Despite this, our understanding of $N_2O_5$ and $ClNO_2$ chemistry in highly polluted urban regions with high levels of $NO_x$ and $O_3$, and high particulate matter is far from complete.

Beijing has been suffering from severe haze pollution during the last two decades (Chan and Yao, 2008). As a result, extensive studies have been conducted to characterize the sources and formation mechanisms of haze episodes (Guo et al., 2014; Huang et al., 2014; Li et al., 2017). The results show that nitrate and its precursors have been playing increasingly important roles in pollution events since 2006 mainly due to the continuous decrease in $SO_2$ (van der A et al., 2017). While the formation mechanisms of nitrate are relatively well known, the relative contributions of different mechanisms can have large variability and uncertainties. Pathak et al. (2009) found that heterogeneous hydrolysis of $N_2O_5$ contributed 50-100% of the nighttime enhancement of nitrate concentration in Beijing. However, WRF-Chem model simulations showed only 21% enhancement of nitrate during highly polluted days (Su et al., 2016). A recent study also observed a large nocturnal nitrate formation potential from $N_2O_5$ heterogeneous uptake, which is comparable to and even higher than that from the partitioning of $HNO_3$ in rural Beijing in autumn (Wang et al., 2017a). A large contribution of heterogeneous hydrolysis of $N_2O_5$ to the high $PM_{2.5}$ nitrate even in the daytime, due to persistently high $NO_2$, was also reported in Hong Kong (Xue et al., 2014). All these results highlight that $N_2O_5$ heterogeneous uptake might be an important pathway of nitrate formation in Beijing. However, the roles of $N_2O_5$ in nitrate formation, and of $N_2O_5$ and $ClNO_2$ in night- and day-time chemistry in summer in urban Beijing are not characterized yet, except for one measurement in suburban Beijing in the summer of 2016 (Wang et al., 2018).

In this work, two high-resolution time-of-flight CIMSs using the same iodide ionization system operated by the Institute of Atmospheric Physics (IAP-CIMS) and University of Manchester (UoM-CIMS), respectively, were deployed in urban Beijing for real-time measurements of gas phase $N_2O_5$ and $ClNO_2$. A broadband cavity enhanced absorption spectrometer (BBCEAS) operated by the University of Cambridge was also deployed synchronously for the inter-comparison of $N_2O_5$. The temporal variations of $N_2O_5$ and $ClNO_2$ in summer and their relationships are characterized. The heterogeneous $N_2O_5$ uptake coefficients and $ClNO_2$ production yields are estimated, and their implications in nitrate formation are elucidated.

## 2    Experimental methods

## 2.1 Field campaign site and meteorology

The measurements were conducted during the Air Pollution and Human Health (APHH) summer campaign from 11 to 16 June, 2017 at the Institute of Atmospheric Physics (IAP), Chinese Academy of Sciences (39°58′28″N, 116°22′16″E, ASL: 49 m), which is an urban site located between the north 3$^{rd}$ and 4$^{th}$ ring roads in Beijing. The meteorological variables including wind direction (WD), wind speed (WS), relative humidity (RH), and temperature ($T$) at 15 m and 100 m were obtained from the Beijing 325 m Meteorological Tower (BMT) at the sampling site. The hourly average RH ranged from 12.9% to 82.8%, with an average value of 36.7%, and the hourly average temperature ranged from 17.9$^{o}$C to 38.7$^{o}$C, averaged at 26.7$^{o}$C. All IAP instruments were deployed on the roof of a two-storey building (~10 m) while those of UoM-CIMS and BBCEAS were housed in two containers at ground level (~4 m) which are approximately 20 m away. More details about the sampling site can be found in previous studies (Sun et al., 2012).

## 2.2 Instruments

### 2.2.1 IAP-CIMS

Ambient air was drawn inside the sampling room through a ~2 m Teflon perfluoroalkoxy tubing (PFA, ¼ inch inner diameters) at a flow rate of 10 standard liters per minute (slm), from which ~2 slm was sub-sampled into the CIMS. Methyl iodide gas (CH$_3$I) from a heated CH$_3$I permeation tube cylinder (VICI, 170-015-4600-U50) was ionized by flowing through a soft X-ray ionization source (Tofwerk AG, type P) under an ultra-high purity nitrogen (N$_2$, 99.999%) flow (2.5 slm). This flow enters an ion molecule reaction (IMR) chamber which was maintained at a pressure of 200 mbar using an SH-112 pump fitted with a Tofwerk blue pressure control box to account for changes in ambient pressure. A short segmented quadrupole (SSQ) positioned behind the IMR was held at a pressure of 2 mbar using a Tri scroll 600 pump. Note that the voltage settings used for the guidance of ions were carefully tuned to avoid declustering as much as possible (Lopez-Hilfiker et al., 2016). The gas phase background was determined once during the campaign by passing dry N$_2$ into inlet for 5 minutes.

### 2.2.2 UoM-CIMS

The UoM-CIMS setup has been described elsewhere (Priestley et al., 2018) except a Filter Inlet for Gases and AEROsols

(FIGAERO) (Lopez-Hilfiker et al., 2014) was used in this study. The gas phase inlet of UoM-CIMS consisted of 5 m ¼" I.D. PFA tubing connected to a fast inlet pump with a total flow rate of 13 slm from which the ToF-CIMS sub-sampled 2 slm. CH$_3$I gas mixtures in N$_2$ were made in the field using a custom-made manifold (Bannan et al., 2014). 20 standard cubic centimetres per minute (sccm) of the CH$_3$I mixture was diluted in 4 slm N$_2$ and ionized by flowing through a Tofwerk x-ray

ionization source. This flow enters into the IMR which was maintained at a pressure of 400 mbar using an SSH-112 pump also fitted with a Tofwerk blue pressure control box, while the subsequent SSQ was held at a pressure of 2 mbar using a Tri scroll 600 pump. During the campaign, gas phase backgrounds were established through regularly overflowing the inlet with dry N$_2$ for 5 continuous minutes every 45 minutes as has been performed previously.

  The ambient target molecules were first ionized by reagent ions in the IMR, and then detected as adduction products

with iodide, for instance, ClNO$_2$ as I•ClNO$_2^-$ at $m/z$ 208 and $m/z$ 210 (I•$^{37}$ClNO$_2^-$), and N$_2$O$_5$ as I•N$_2$O$_5^-$ at $m/z$ 235 (Slusher et al., 2004; Kercher et al., 2009) at a time resolution of 1 s. Data analysis is performed using the "Tofware" package (version 2.5.11) running in Igor Pro (WaveMetrics, OR, USA) environment. The mass axis of UoM-CIMS was calibrated using I$^-$, I$_2^-$ and I$_3^-$, while that of IAP-CIMS was calibrated using NO$_3^-$, I$^-$, I•H$_2$O$^-$, I•CH$_2$O$_2^-$, I•HNO$_3^-$, and I$_3^-$, covering a wide range from $m/z$ 62 to 381. Examples of high resolution peak fittings of $m/z$ 208, 210 and 235 for IAP-CIMS are

presented in Fig. S1.

### 2.2.3 Broadband cavity enhanced absorption spectrometer (BBCEAS)

A detailed description of BBCEAS has been given in Kennedy et al. (2011). Briefly, ambient air is first heated to 140 $^{\circ}$C to thermally dissociate N$_2$O$_5$ into NO$_3$ and then enters the observational cavity that consists of two high-reflectivity mirrors. The sum of N$_2$O$_5$ and NO$_3$ is determined using the measured optical absorption of NO$_3$ in the wavelength of 640-680 nm.

The temperature of the cavity is kept at 85±1 $^{\circ}$C to prohibit the recombination of NO$_3$ and NO$_2$ and to maintain the stability of the optical transmission signal. A very fast flow rate of 20 lpm is adopted to minimize the residence time of gases through PFA tubes. The loss of NO$_3$ through the system was estimated to be approximately 10%.

  Considering that the relatively high aerosol loadings in Beijing can attenuate the intracavity light intensity and thus deteriorate instrument sensitivity, a poly tetrafluoroethylene (PTFE) filter of pore size 1 μm was used to remove aerosol



particles from the air stream. This filter acts also a point loss (~10%) for $NO_3$ but has a negligible impact on $N_2O_5$ (Dube et al., 2006). Because the mixing ratio of $N_2O_5$ is higher than $NO_3$ by a factor of >10 during APHH summer campaign, the influence of filter loss on the measurements of $N_2O_5+NO_3$ is expected to be small. Aging of aerosol particles on the filter may potentially introduce uncertainties for the transmission efficiencies of $NO_3$ and $N_2O_5$, but was found to be insignificant in this study.

## 2.3 Calibrations and inter-comparisons

During the campaign, field calibrations for UoM-CIMS were regularly carried out using known concentration formic acid gas mixtures made in the custom-made manifold. A range of other species were calibrated after the campaign, and relative calibration factors were derived using the measured formic acid sensitivity during these calibrations as has been performed previously (Le Breton et al., 2014, 2017; Bannan et al., 2014, 2015).

The UoM-CIMS was calibrated post campaign for both $N_2O_5$ and $ClNO_2$, relative to formic acid that was calibrated and measured throughout the campaign. This is completed assuming that the ratio between formic acid and $ClNO_2$ sensitivity remains constant. $ClNO_2$ was calibrated using the method described in Kercher et al. (2009). Briefly, a stable source of $N_2O_5$ is generated and passed over a salt slurry where excess chloride reacts to produce gaseous $ClNO_2$. The $N_2O_5$ for this process was synthesised based on the methodology described by Le Breton et al. (2014). Excess $O_3$ is generated through flowing 200 sccm $O_2$ (BOC) through an ozone generator (BMT, 802N) into a 5 litre glass volume containing $NO_2$ (Sigma, >99.5%). The outflow from this reaction vessel is cooled in a cold trap held at -78°C (195K) by a dry ice/glycerol mixture where $N_2O_5$ is condensed and frozen. The trap is allowed to reach room temperature and the flow is reversed where it is then condensed in a second trap held at 220 K. This process is repeated several times to purify the mixture. The system is first purged by flowing $O_3$ for ten minutes before usage. To ascertain the $N_2O_5$ concentration in the line, the flow is diverted through heated line to decompose the $N_2O_5$ and into to a Thermo Scientific 42i $NO_x$ analyser where it is detected as $NO_2$. According to the inter-comparisons with the BBCEAS, including this study and others (e.g., Le Breton et al. (2014); Bannan et al. (2017)), the possible interference of $NO_y$ on the $NO_x$ analyser is not deemed important in terms of our reported $N_2O_5$ concentrations.

$ClNO_2$ was produced by flowing a known concentration of $N_2O_5$ in dry $N_2$ through a wetted NaCl scrubber. Conversion

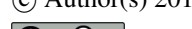



of $N_2O_5$ to $ClNO_2$ can be as efficient as 100% on sea salt, but it can also be lower, for example if $ClNO_2$ is converted to $Cl_2$ (Roberts et al., 2008). In this calibration we have followed the accepted methods of Osthoff et al. (2008) and Kercher et al. (2009) that show a conversion yield of 100% and have assumed this yield in the calibrations of this study.

The second method used to verify our $ClNO_2$ calibration is by cross calibration with a turbulent flow tube chemical

ionisation mass spectrometer (TF-CIMS) (Leather et al., 2012). A known concentration of 0-20 sccm $Cl_2$ (99.5% purity $Cl_2$ cylinder, Aldrich) from a diluted (in $N_2$) gas mix is flowed into an excess constant flow of 20 sccm $NO_2$ (99.5% purity $NO_2$ cylinder, Aldrich) from a diluted (in $N_2$) gas mix, to which the TF-CIMS has been calibrated. This flow is carried in 52 slm $N_2$ that is purified by flowing through two heated molecular sieve traps. This flow is sub-sampled by the ToF-CIMS where the I•$ClNO_2^-$ adduct is observed. The TF-CIMS is able to quantify the concentration of $ClNO_2$ generated in the flow tube as

the equivalent drop in $NO_2^-$ signal. This indirect measurement of $ClNO_2$ is similar in its methodology to $ClNO_2$ calibration by quantifying the loss of $N_2O_5$ reacted with $Cl^-$ (e.g., Kercher et al. (2009)). The TF-CIMS method gives a calibration factor 58% greater than that of the $N_2O_5$ synthesis method therefore this is taken as our measurement uncertainty. This calibration was scaled to those in the field using formic acid calibrations carried out in the laboratory by overflowing the inlet with various known concentrations of gas mixtures (Bannan et al., 2014).

The IAP-CIMS calibration for $N_2O_5$ was performed by comparing with the measurements from the BBCEAS. As shown in Fig. S2, the raw signals of $N_2O_5$ from the IAP-CIMS measurements were highly correlated with those from BBCEAS ($R^2 = 0.84$). Given that the inter-comparison between the two instruments was relatively constant throughout the study, the average regression slope of 0.54 was then applied to determine the mixing ratio of $N_2O_5$ for the IAP-CIMS. The estimated $N_2O_5$ mixing ratios were then compared with those measured by UoM-CIMS. As shown in Fig. 1, the two $N_2O_5$

measurements tracked well with each other ($R^2 = 0.84$, slope =1.42) although some differences at the midnight of 13 June were observed. The raw signals of $ClNO_2$ given by the IAP-CIMS were first converted to mixing ratios by assuming the same sensitivity between $ClNO_2$ and $N_2O_5$ (i.e., 0.54 cps pptv$^{-1}$). The results show that the estimated $ClNO_2$ for the IAP-CIMS agrees well with that was measured by UoM-CIMS and calibrated post campaign ($R^2 = 0.93$, slope = 0.905, Fig. 1). All the discussions below are based on IAP-CIMS measurements unless otherwise stated.



## 2.3 Collocated measurements

An Aerodyne High-Resolution Time-of-Flight Aerosol Mass Spectrometer (HR-AMS hereafter) and an Aethalometer (AE33, Magee Scientific Corp.) were deployed on the roof of the two-storey building to measure the size-resolved non-refractory submicron aerosol (NR-PM$_1$) species with a time resolution of 5 min, including organics (Org), sulfate (SO$_4^{2-}$), nitrate

(NO$_3^-$), ammonium (NH$_4^+$), and chloride (Cl$^-$) (DeCarlo et al., 2006; Canagaratna et al., 2007), and black carbon (BC), respectively. A more detailed description of the operations and calibrations of this HR-AMS can be found in Xu et al. (2015) and Sun et al. (2016). Other collocated measurements in two containers at ground level included gaseous species of O$_3$ (TEI 49C UV absorption analyzer), NO (TEI 42i TL NO analyzer), and NO$_2$ (CAPS NO$_2$ monitor, Aerodyne Research Inc.), and size-resolved particle number concentrations (11-550 nm) by a scanning mobility particle sizer (SMPS) equipped with a long

Differential Mobility Analyzer (DMA, TSI, 3081A) and a Condensation Particle Counter (CPC, TSI, 3772).

## 2.4 Data analysis

### 2.4.1 Estimation of $\gamma_{\mathrm{N_2O_5}}$ and ø

NO$_3$ is formed from the reaction of NO$_2$ with O$_3$ (R1) with a temperature-dependent reaction rate constant $k_1$. NO$_3$ rapidly photolyzes during daytime, but at night it reacts with NO$_2$ to produce N$_2$O$_5$ (R2). N$_2$O$_5$ can thermally decompose back to

NO$_3$ and NO$_2$, and the equilibrium rate coefficient $K_{\mathrm{eq}}$ is a function of ambient temperature. In this study, values of $k_1$ and $K_{\mathrm{eq}}$ recommended by Atkinson et al. (2004) and Brown and Stutz (2012) were used. The indirect loss of N$_2$O$_5$ is mainly through the reactions of NO$_3$ with either NO or VOCs (R3), while the direct N$_2$O$_5$ loss is predominantly from the heterogeneous hydrolysis on the surface of aerosol particles that contain water (R4) or chloride (R5). Note that het is an abbreviation of heterogeneous in the equations. The net reaction of R4 and R5 can be described as R6 where $k_{\mathrm{N_2O_5}}$ is the

heterogeneous uptake rate coefficient for N$_2$O$_5$, and ø is the ClNO$_2$ yield.

$$\mathrm{NO_2 + O_3 \rightarrow NO_3 + O_2},\ k_1 \tag{R1}$$

$$\mathrm{NO_2 + NO_3 + M \leftrightarrow N_2O_5 + M},\ K_{\mathrm{eq}} \tag{R2}$$

$$\mathrm{NO_3 + (NO\ or\ VOCs) \rightarrow products},\ k_{\mathrm{NO_3}} \tag{R3}$$

$$\mathrm{N_2O_5 + H_2O(het) \rightarrow 2HNO_3(aq)} \tag{R4}$$



$$N_2O_5 + Cl^-(het) \rightarrow NO_3^-(aq) + ClNO_2 \tag{R5}$$

$$N_2O_5 + (H_2O \text{ or } Cl^-)(het) \rightarrow (2\text{-ø})NO_3^-(aq) + ø \ ClNO_2, \ k_{N_2O_5} \tag{R6}$$

When the uptake reaction was not limited by gas-phase diffusion, $k_{N_2O_5}$ can be simplified as Eq. (1) (Dentener and Crutzen, 1993; Riemer et al., 2003):

$$k_{N_2O_5} = \frac{1}{4} \times c \times S_a \times \gamma_{N_2O_5} \tag{1}$$

Where $c$ is the mean molecular speed of $N_2O_5$ (unit, m s$^{-1}$), and $S_a$ is the aerosol surface area density calculated from the size-resolved particle number concentrations assuming spherical particles (unit, μm cm$^{-3}$). Note that $S_a$ determined under dry conditions was converted to that under ambient RH levels by using the hygroscopic growth factor in Liu et al. (2013).

The nocturnal mixing ratio of $NO_3$ can be derived from the simultaneous measurements of $NO_2$ and $N_2O_5$ (R2) assuming that the equilibrium between $NO_3$ and $N_2O_5$ is rapidly established after sunset (Brown et al., 2003a).

$$[NO_3(cal)] = \frac{[N_2O_5]}{K_{eq}[NO_2]} \tag{2}$$

The nitrate radical production rate $p(NO_3)$ can be calculated from reaction R1 assuming that the nitrate radical is solely from reaction R1.

$$p(NO_3) = k_1[NO_2][O_3] \tag{3}$$

With a steady-state assumption for $NO_3$ and $N_2O_5$, the inverse $N_2O_5$ steady state lifetime, $\tau(N_2O_5)^{-1}$, which is defined as the ratio of $p(NO_3)$ to the $N_2O_5$ mixing ratios, can be expanded to Eq. (4) after the substitution of Eqs. (2) and (3) into the approximate time change rate for $N_2O_5$ (Brown et al., 2003a).

$$\tau(N_2O_5)^{-1} = \frac{p(NO_3)}{[N_2O_5]} \approx \frac{k_{NO_3}}{K_{eq}[NO_2]} + k_{N_2O_5} \tag{4}$$

$\frac{k_{NO_3}}{K_{eq}[NO_2]}$ represents the contribution to $\tau(N_2O_5)^{-1}$ from the indirect $N_2O_5$ loss pathway, i.e. through the $NO_3$ reactions with VOCs and/or NO, while $k_{N_2O_5}$ indicates the direct loss of $N_2O_5$ through heterogeneous uptake.

Considering that the production of $ClNO_2$ is predominantly from the heterogeneous $N_2O_5$ uptake within stable air masses and precursors, the production rate of $ClNO_2$ ($pClNO_2$) can be related to the heterogeneous loss rate of $N_2O_5$ by:

$$pClNO_2 = \frac{dClNO_2}{dt} = ø \times (\frac{1}{4} \times c \times S_a \times \gamma_{N_2O_5}) \tag{5}$$



The production rate of particulate nitrate ($p\text{NO}_3^-$) was obtained from HR-AMS measurements assuming that the measured $p\text{NO}_3^-$ can account for the total production of nitrate by reaction R4 (Phillips et al., 2016).

$$p\text{NO}_3^- = \frac{d\text{NO}_3^-}{dt} = (2-\o) \times (\frac{1}{4} \times c \times S_a \times \gamma_{\text{N}_2\text{O}_5}) \qquad (6)$$

Only periods with concurrent formation of $\text{ClNO}_2$ and $\text{NO}_3^-$ meet the requirements that both of them are produced only from the heterogeneous $\text{N}_2\text{O}_5$ uptake. By combining Eq. (5) with Eq. (6), $\gamma_{\text{N}_2\text{O}_5}$ and $\o$ can be represented as:

$$\gamma_{\text{N}_2\text{O}_5} = \frac{2(p\text{ClNO}_2 + p\text{NO}_3^-)}{c \times S_a \times [\text{N}_2\text{O}_5]} \qquad (7)$$

$$\o = 2 \left(\frac{p\text{NO}_3^-}{p\text{ClNO}_2} + 1\right)^{-1} \qquad (8)$$

### 2.4.2 Parameterization of $\gamma_{\text{N}_2\text{O}_5}$ and $\o$

Aerosol liquid water content associated with inorganic species was estimated using the ISORROPIA-II thermodynamic equilibrium model (Nenes et al., 1998; Fountoukis and Nenes, 2007), with input data of ambient NR-PM$_1$ species, and RH and $T$ at 15 m. The $\text{N}_2\text{O}_5$ uptake coefficient and $\text{ClNO}_2$ yield can also be calculated by the parameterization proposed by Bertram and Thornton (2009).

$$\gamma_{\text{N}_2\text{O}_5} = Ak(1- \frac{1}{1+\frac{29[Cl^-]}{[\text{NO}_3^-]}+\frac{0.06[\text{H}_2\text{O}]}{[\text{NO}_3^-]}}) \qquad (9)$$

$$\o = (1+\frac{[\text{H}_2\text{O}]}{483[Cl^-]})^{-1} \qquad (10)$$

Where [$\text{H}_2\text{O}$], [$\text{NO}_3^-$] and [$Cl^-$] are molar concentrations of liquid water, particle nitrate and chloride, respectively, and the empirical parameters $A = 3.2 \times 10^{-8}$, and $k = 1.15 \times 10^6 \times (1- e^{-0.13[\text{H}_2\text{O}]})$ are used.

## 3 Results and discussion

### 3.1 Overview of $\text{N}_2\text{O}_5$ and $\text{ClNO}_2$ measurements

Figure 1 shows the time series of $\text{N}_2\text{O}_5$, $\text{ClNO}_2$ and $p(\text{NO}_3)$, submicron aerosol species of $\text{NO}_3^-$ and non-refractory $\text{Cl}^-$, gaseous species of NO, $\text{NO}_2$ and $\text{O}_3$, and meteorological parameters during the field campaign. Both $\text{N}_2\text{O}_5$ and $\text{ClNO}_2$ exhibited large day-to-day variability with the average ($\pm 1\sigma$) mixing ratios being $79.2 \pm 157.1$ pptv and $174.3 \pm 262.0$ pptv,



respectively. Such dramatic variations of $N_2O_5$ and $ClNO_2$ are consistent with previous observations in various environments, for example, ground sites in Colorado and London (Thornton et al., 2010; Bannan et al., 2015) and the residual layer in Mt. Tai (Wang et al., 2017c). Four nights (i.e., P1, P2, P3 and P4 from 20:00 to 04:30) were selected to investigate nocturnal chemistry of $N_2O_5$ and $ClNO_2$ in this study. The first two nights (P1 and P2) showed much higher mixing ratios of $N_2O_5$ and

$ClNO_2$ than those during P3 and P4, although the $NO_x$ and $O_3$ levels during P4 were comparable to those during P2 (Table 1).

The highest $N_2O_5$ mixing ratio (1.10 ppbv, 1-minute average) was observed at 2:15 on 13 June (P2), which is comparable to the previous observation in urban Beijing (1.3 ppbv) (Wang et al., 2017a), but much lower than that in the aged air masses in Hong Kong ~7.8 ppbv (Brown et al., 2016a). A recent measurement at a suburban site in Beijing impacted by the outflow of urban Beijing air masses also reported consistently high $N_2O_5$ (1-minute maxima 937 pptv) (Wang et al.,

2018). The mixing ratio of $N_2O_5$ was also much higher than that in the nocturnal residential boundary layer in Mt. Tai (167 pptv) (Wang et al., 2017c), indicating potentially significant nighttime $N_2O_5$ chemistry in highly polluted urban areas. One of the reasons for this could be due to the high mixing ratios of precursors, for instance, the average $O_3$ mixing ratios at night-time were as high as 18-56 ppbv. The lowest nighttime average of $N_2O_5$ (~ 37.8 pptv) was observed during P3 mainly attributed to the fast heterogeneous hydrolysis of $N_2O_5$ under high RH (~60.5%) conditions.

Similar to $N_2O_5$, $ClNO_2$ presented the highest value (1.44 ppbv, 1-minute average) before sunrise on 13 June (P2), yet it is lower than the maximum of 2.1 ppbv (1-minute average) observed at a rural site located to the southwest of Beijing (Tham et al., 2016), and also the $ClNO_2$ peak of 2.9 ppbv (1-minute average) in suburban Beijing (Wang et al., 2018). These results indicated ubiquitously observed $ClNO_2$ in the NCP, although high $ClNO_2$ mixing ratios have been also observed previously in both marine and continental environments in North America, Europe and Asia (Osthoff et al., 2008; Thornton et al., 2010;

Mielke et al., 2011; Phillips et al., 2012; Tham et al., 2014). The average nitrate radical production rate $p(NO_3)$ was 2.8 and 3.6 during P1 and P2, respectively, which are both higher than those during P3 and P4 (1.7-2.6) (Table 1). This result supports a higher production potential for $N_2O_5$ during P1 and P2. On average, $p(NO_3)$ was $2.6 \pm 2.4$ ppbv h$^{-1}$ at night-time, indicating more active nocturnal chemistry than previous studies in NCP in terms of radical production rates, for example, $1.2 \pm 0.9$ ppbv h$^{-1}$ in suburban Beijing, $1.7 \pm 0.6$ ppbv h$^{-1}$ in Wangdu, and $0.45 \pm 0.40$ ppb h$^{-1}$ in Mt. Tai (Tham et al., 2016;

Wang et al., 2017c; Wang et al., 2018). We also note that the $p(NO_3)$ was comparable between P4 and P2 (2.6 pptv vs. 2.8



pptv), yet the $N_2O_5$ and $ClNO_2$ mixing ratios during P4 were much lower, likely due to the favorable dispersing meteorological conditions with higher wind speed and lower relative humidity (Table 1). Our results illustrate that precursors levels, reaction rates, and meteorological conditions can all affect the variability of $N_2O_5$ and $ClNO_2$.

The average diurnal variations of trace gases, $N_2O_5$, $ClNO_2$ and submicron nitrate and chloride are depicted in Fig. 2. $O_3$
showed a pronounced peak of 93.3 ppbv between 14:00 and 16:00 corresponding to a minimum mixing ratio of $NO_2$ (9.1 ppbv). As a consequence, $p(NO_3)$ showed relatively high values around noon with a decrease in the middle of the afternoon owing to the depletion of $NO_2$ and then reached a maximum of 5.9 ppbv $h^{-1}$ before sunset. A similar diurnal pattern of $p(NO_3)$ was also observed at a rural site in the autumn of Beijing (Wang et al., 2017a). Both NO and $NO_2$ showed pronounced diurnal cycles with lowest concentrations in the afternoon. In addition to the rising boundary layer, the formation of $NO_z$ is
another important reason for the low levels of $NO_x$ during this time period in urban Beijing (Sun et al., 2011). Nitrate and chloride also showed lowest concentrations in the late afternoon, mainly due to the evaporative loss under high temperature conditions (Sun et al., 2012).

After sunset, $N_2O_5$ was rapidly formed associated with a corresponding decrease in $p(NO_3)$. The mixing ratio of $N_2O_5$ peaked approximately at 22:00 and then remained at a consistently high level (~200-300 pptv) until 3:00. After that, $N_2O_5$
showed a rapid decrease due to the significant titration by NO. Similar loss of $N_2O_5$ due to the injection of NO-containing air was also reported at sites near urban areas (Brown et al., 2003b). Because NO is predominantly from local emissions as supported by the tight correlation ($R^2 = 0.64$-$0.73$, Fig. S3) with black carbon, a tracer for combustion emissions, our results demonstrated that the local NO emissions serve as the most important scavenger of $N_2O_5$ before sunrise in urban Beijing. In comparison, the decrease in $N_2O_5$ due to the NO titration only occurred during the second half of the night with low $O_3$ in
suburban Beijing (Wang et al., 2018). This study also found high $N_2O_5$ after midnight due to the incomplete titration of $O_3$, for instance, ~52.9 ppbv after midnight on 13 June, which is different from previous findings that high $N_2O_5$ mixing ratios were typically observed before midnight due to the rapid depletion of $O_3$ (Wang et al., 2017a; Wang et al., 2017c). The high nocturnal mixing ratios of $O_3$ and $NO_2$ (Fig. 2) highlight much higher oxidative capacity at night in summer in urban Beijing compared to the other seasons and/or rural locations.
$ClNO_2$ showed clear nocturnal formation from heterogeneous processing and decreased rapidly after sunrise mainly due



to photolysis (Fig. 2). Note that ClNO$_2$ peaked at a similar time (21:00-22:00) as that of N$_2$O$_5$ without showing a time lag of 1-3 h as previously observed in Jinan (Wang et al., 2017b), indicating that either particulate Cl$^-$ was sufficient for the heterogeneous reactions or other chlorine sources (e.g., HCl) contributed to the formation of ClNO$_2$ in urban Beijing (Riedel et al., 2012). According to previous studies, the partitioning of HCl to particulate Cl$^-$ could contribute to ClNO$_2$ formation

substantially at urban sites (Thornton et al., 2010). In addition, Wang et al. (2018) also speculated that large particle chloride during the campaign was possibly replenished by gas-phase HCl due to the high emissions of from human activities. We also found that ClNO$_2$ was well correlated with chlorine (Cl$_2$) derived from IAP-CIMS (R$^2$ = 0.90-0.99) rather than particulate chloride (Cl$^-$) (R$^2$ = 0.01-0.44) at the night-time, indicating that ClNO$_2$ might act as an intermediate during the formation of Cl$_2$ under sufficient chloride conditions (Roberts et al., 2008). Indeed, the much lower particulate Cl$^-$ than ClNO$_2$ also

indicated other chlorine sources. Therefore, we need simultaneous measurements for further supporting this conclusion in this study, e.g., HCl.

## 3.2    Reactivity of N$_2$O$_5$ and NO$_3$

Considering the time needed for meeting the steady-state assumption, only the two-hour data after sunset were used to calculate N$_2$O$_5$ steady-state lifetime via Eq. (4) (Wagner et al., 2013). High N$_2$O$_5$ reactivity was observed during these four

nights with average $\tau$(N$_2$O$_5$)$^{-1}$ ranging from $0.16\times10^{-2}$ to $1.58\times10^{-2}$ s$^{-1}$, which corresponds to a nighttime N$_2$O$_5$ lifetime between 1.1 and 10.7 min (Fig. 3). Such values are overall consistent with those measured at surface sites and in the nocturnal residual layer in NCP, for example, $1.30\times10^{-2}$ s$^{-1}$ in Wangdu (Tham et al., 2016) and $1.30$-$1.40\times10^{-2}$ s$^{-1}$ in Mt. Tai (Wang et al., 2017c). In comparison, the N$_2$O$_5$ loss is much more rapid than that previously reported in southern China (Brown et al., 2016a) and the USA (Brown et al., 2009; Wagner et al., 2013), mainly due to the high aerosol loading in NCP

leading to an enhanced N$_2$O$_5$ sink through both indirect and direct pathways. Correspondingly, the average $\tau$(NO$_3$)$^{-1}$ calculated from the inferred NO$_3$ were $0.02$-$0.62$ s$^{-1}$ during the four nights, indicating active NO$_3$ nighttime chemistry through reactions with NO and VOCs in the polluted nocturnal boundary. Note that P2 and P4 showed comparable $p$(NO$_3$) (2.8 vs. 2.6 ppbv h$^{-1}$) (Table 1), yet the N$_2$O$_5$ reactivity during P4 ($1.58\times10^{-2}$ s$^{-1}$) was significantly higher than that during P2 ($0.16\times10^{-2}$ s$^{-1}$) likely due to enhanced N$_2$O$_5$ heterogeneous loss (discussed in Sec.3.4). Consistently, $\tau$(NO$_3$)$^{-1}$ showed similar



patterns to those of $\tau(N_2O_5)^{-1}$. Indeed, the $N_2O_5$ reactivity presented a nonlinear dependence on aerosol surface area ($S_a$) and relative humidity (Figs. 3c and 3d). Although P3 showed much higher RH than P4 (60.5% vs. 28.0%), the $N_2O_5$ reactivity was comparable between P3 and P4 (0.014 vs. 0.016 s$^{-1}$), illustrating the complex heterogeneous process of $N_2O_5$.

Figure 3c shows the $N_2O_5$ lifetime as a function of surface area density. $\tau(N_2O_5)$ decreased rapidly from 11.8 minutes to

2.2 minutes as $S_a$ increased up to 500 $\mu m^2\, cm^{-3}$ , and then remained at relatively constant levels at $S_a > 500$ $\mu m^2\, cm^{-3}$. Such an $S_a$ dependence of $\tau(N_2O_5)$ is consistent with previous observations in Hong Kong (Brown et al., 2016a). Large variations in $\tau(N_2O_5)$ as a function of RH were also observed. As shown in Fig. 3d, the $N_2O_5$ lifetime decreased by nearly a factor of 5 from 11.3 to 2.2 mins as RH increased from 40% to 50%. We noticed that the aerosol surface area exhibits an increase as a function of RH at RH > 40% (Fig. S4). These results suggested that the decrease in $\tau(N_2O_5)$ at high RH levels (RH >40%)

might be caused by increased $N_2O_5$ uptake rates due to the higher $S_a$. In addition, the increasing aerosol liquid water content at high RH might be another reason (Fig. S4). Comparatively, the $N_2O_5$ lifetime showed an increase as a function of RH at RH< 40%, while the variations in $S_a$ were small, suggesting additional contributions from other factors, for example, aerosol loading and composition (Morgan et al., 2015). Considering that the period of this study is relatively short, long-term measurements are needed in future studies to better characterize the parameterizations of $\tau(N_2O_5)$ as a function of $S_a$ and RH.

**3.3   Relationship between $N_2O_5$ and $ClNO_2$**

Previous studies have found that $N_2O_5$ and $ClNO_2$ were generally positively correlated in predominantly continental air masses whereas they were negatively correlated in marine air masses with high chloride content (Bannan et al., 2015). Phillips et al. (2012) also reported large variability in $N_2O_5$ and $ClNO_2$ correlations and $ClNO_2$-to-$N_2O_5$ ratios in air masses from continental or marine origins due to the changes in particle Cl$^-$. In this study, $ClNO_2$ was well and positively correlated

with $N_2O_5$ during all four nights (Fig. 4, $R^2$ = 0.60-0.88), and only slight changes in $ClNO_2/N_2O_5$ ratios were observed after sunset. These results are different from previous observations showing large variability in the correlations (Osthoff et al., 2008), which indicates that particulate Cl$^-$ was always sufficient for the $ClNO_2$ formation during this study period. The differences in regression slopes among the four nights can be explained by different air masses originating from different regions which were calculated using the Hybrid Single Particle Lagrangian Integrated Trajectory (HYSPLIT, NOAA) model



(Draxler and Hess, 1997) (Fig. S5). For example, $ClNO_2$ tracked much better with $N_2O_5$ after midnight ($R^2 = 0.69$) than that before midnight ($R^2 = 0.16$) during P2 (Fig. S6), suggesting the influences of air masses from different regions (Fig. S5). Comparatively, P4 and P1 showed similar tight correlations between $ClNO_2$ and $N_2O_5$ before and after midnight, consistent with their similar back trajectories during the two different periods.

The $ClNO_2/N_2O_5$ ratios varied significantly throughout the study ranging from 0.3 to 95.5 (30 minute average). The average ($\pm 1\sigma$) ratio of $ClNO_2/N_2O_5$ was $6.9 \pm 7.4$, consistent with the previous studies in NCP, for example, 0.4-131.3 in Jinan and Wangdu (Tham et al., 2016; Wang et al., 2017b). However, the ratios are substantially higher than those measured in other megacities, e.g., Hong Kong (0.1-2.0) (Wang et al., 2016b), London (0.02-2.4) (Bannan et al., 2015) and Los Angeles, California (0.2-10.0) (Mielke et al., 2013). These results indicate ubiquitously high $ClNO_2/N_2O_5$ ratios in the NCP,

consistent with another measurement in suburban Beijing (Wang et al., 2018), which might result from the high $ClNO_2$ production rate due to high aerosol loadings. We also note that the relatively low $N_2O_5$ associated with high $N_2O_5$ reactivity might be another possible explanation. Furthermore, we compared the $ClNO_2/N_2O_5$ ratios with particulate concentrations and compositions during the four nights (Fig. 5). P3 showed the highest median ratio of 9.4, which is much higher than those during the rest of three nights (1.0-3.2). This can be explained by the correspondingly high liquid water content that

facilitated the $N_2O_5$ heterogeneous uptake (Morgan et al., 2015). In comparison, the particle chloride concentrations were relatively close during the four nights , with slightly lower concentrations during P4, further supporting that the $ClNO_2/N_2O_5$ ratios were independent of particle chloride in this study due to the sufficient chloride source for the $ClNO_2$ production, e.g., HCl gas-particle partitioning. The lower $ClNO_2/N_2O_5$ ratios during P2 compared with P1 can be explained by the "nitrate effect" which suppressed $N_2O_5$ uptake (Mentel and Wahner, 1999) as P2 showed much higher nitrate concentrations than P1

(4.2 vs. 1.4 $\mu g\ m^{-3}$). Note that the $ClNO_2/N_2O_5$ ratios were also characterized with the dependence on $Org/SO_4$ ratios in our campaign, similar to other studies (Evans and Jacob, 2005; Riemer et al., 2009).

## 3.4   $N_2O_5$ uptake coefficient and $ClNO_2$ production yield

To quantity the relative contributions of different pathways to $N_2O_5$ loss, three periods with relatively stable air masses and concurrent increases in $ClNO_2$ and $NO_3^-$ (Fig. 6, 20:00-23:00 on 12 June, 22:00-00:00 on 13 June, and 20:00-22:30 on





14 June) were selected for the calculations of $\gamma_{N_2O_5}$ and ø. The rigorous method as suggested by Phillips et al. (2016) was used in this study. Briefly, the predicted concentrations of $ClNO_2$ and $NO_3^-$ were derived by integrating $pClNO_2$ and $pNO_3^-$ with average $S_a$ and $N_2O_5$ over each time step (~15 min) and initial estimations for $\gamma_{N_2O_5}$ and ø. Repeating the integration by changing $\gamma_{N_2O_5}$ and ø until good agreements between observed and predicted values of $ClNO_2$ and $NO_3^-$ were reached. The

derived heterogeneous uptake coefficient, $ClNO_2$ yield, and $N_2O_5$ loss rate $k_d$ following this method are listed in Table 2.

The estimated $\gamma_{N_2O_5}$ for the three selected periods were 0.017-0.09, which was generally comparable to previous values (0.014-0.092) derived from the steady-state assumption method in the NCP (Tham et al., 2016; Wang et al., 2017a; Wang et al., 2017b; Wang et al., 2017c), and also consistent with the recent measurements (0.012-0.055) using the same method in suburban Beijing (Wang et al., 2018). However, the $\gamma_{N_2O_5}$ determined in our campaign was 1-2 orders of

magnitude higher than those obtained in laboratory (Thornton et al., 2003), and also much higher than those in Hong Kong and Germany (Brown et al., 2016a; Phillips et al., 2016). We also found that the parameterized $\gamma_{N_2O_5}$ values (0.0014-0.012) determined from Eq. (9) (the Bertram-Thornton parameterization) were significantly lower than the observed values, suggesting that more field measurements are needed to improve the parameterization schemes. Note that $\gamma_{N_2O_5}$ values appeared to increase with the rising relative humidity, which were also observed at other sites (Thornton et al., 2003; Wang

et al., 2017b). For example, $\gamma_{N_2O_5}$ values increased from 0.019 to 0.090 when RH increased from 21.1% to 63.6%. However, the $\gamma_{N_2O_5}$ values were comparable at low RH levels (< 40%) (0.019 vs. 0.017 in Table 2) although RH differed by a factor of 2 (21% vs. 40%). These results further supported that the influences of hygroscopic growth on $\gamma_{N_2O_5}$ were mainly caused by increasing aerosol liquid water content. The direct $N_2O_5$ loss rates estimated from the uptake coefficient were in the range of 0.00044-0.0034 $s^{-1}$, which contributed 6.9-32.6% to the total $N_2O_5$ loss with the rest being indirect loss. Our results

indicated that the fast $N_2O_5$ loss in the nocturnal boundary in urban Beijing was predominantly from the indirect loss of $NO_3$ rather than the heterogeneous uptake of $N_2O_5$, mainly due to active $NO_3$ reaction in summer. Such a conclusion was different from previous results in autumn Beijing that $N_2O_5$ loss was dominated by $N_2O_5$ heterogeneous hydrolysis (69.1%-98.8%) (Wang et al., 2017a). Several studies also revealed the importance of heterogeneous $N_2O_5$ uptake in $N_2O_5$ loss in the NCP by using the steady-state derived $\gamma_{N_2O_5}$ (Tham et al., 2016; Wang et al., 2017b; Wang et al., 2017c). While the uncertainties in

different analysis methods, e.g., the product formation rates or steady-state assumption are one of the reasons, the high VOCs



emissions, particularly biogenic emissions (e.g., isoprene and terpene) in summer than other seasons could be another important reason for the differences in dominant $N_2O_5$ loss pathway. Indeed, the indirect $N_2O_5$ loss via $NO_3$+VOCs was also found to dominate the total loss of $N_2O_5$ (67%) in summer in suburban Beijing (Wang et al., 2018). Our results highlight the significant nighttime $NO_x$ loss through the reactions of $NO_3$ with VOCs in summer in urban Beijing.

The $ClNO_2$ yields ø derived for the three cases were 0.35, 0.10 and 0.15, respectively. The production yields in this study are substantially larger than those in urban Jinan (0.014-0.082) (Wang et al., 2017b), yet comparable to those reported in Mt. Tai (0.02-0.90) (Wang et al., 2017c) and continental Colorado (0.07-0.36) (Thornton et al., 2010). However, the significantly lower ø than that in suburban Beiing (0.50-1.0; Wang et al., 2018) indicated more effective $ClNO_2$ production in suburban regions than urban regions to some extent. Indeed, the product of $\gamma_{N_2O_5}$ and ø ($\gamma_{N_2O_5} \times ø$) in this study ranged

from 0.006-0.009 and was much lower than those in Wang et al. (2018) (0.008-0.035). We noticed that ø were much lower than those paramterized from Eq. (10) (0.55-0.97), indicating that the Bertram-Thornton parameterization scheme might overestimate the $ClNO_2$ yield substantially. Note that $\gamma_{N_2O_5}$ might be overestimated associated with an underestimation of ø if assuming particulate nitrate is completely from the $N_2O_5$ heterogeneous uptake. Possible contribution from gas-phase $HNO_3$ repartitioning to the particulate phase was not considered mainly due to the lack of observational data for $HNO_3$ and

$NH_3$. Indeed, a recent study found that the nocturnal nitrate formation potential by $N_2O_5$ heterogeneous uptake was comparable to that formed by gas-phase $HNO_3$ repartitioning in Beijing (Wang et al., 2017a). In addtion, $\gamma_{N_2O_5} \times ø$ was higher on 13 June than the other two days (e.g., 0.009 vs. 0.003-0.006), which might explain the corresondingly higher $ClNO_2/N_2O_5$ ratio in this day (on average 8.2 vs. 1.2-1.4). Our results overall suggest fast heterogeneous $N_2O_5$ uptake and high $ClNO_2$ production rate in summer in urban Beijing, which might have great implications for models to improve the

simulations for nocturnal nitrate and daytime ozone.

**4    Conclusions**

We present the simultaneous measurement of gas-phase $N_2O_5$ and $ClNO_2$ by I-CIMS during the APHH summer campaign to investigate the nocturnal chemistry in urban Beijing. The average (±1σ) mixing ratios of $N_2O_5$ and $ClNO_2$ were 79.2 ± 157.1 pptv and 174.3 ± 262.0 pptv, with maximum values of 1.17 ppbv and 1.44 ppbv, respectively. Differing from previous





studies with negligible $N_2O_5$ after midnight at surface level, our measurements showed high nocturnal levels of $N_2O_5$ across the entire night, suggesting a high oxidative capacity in summer in urban Beijing. $N_2O_5$ and $ClNO_2$ exhibited clear diurnal variations with significant nocturnal formation due to heterogeneous uptake. The average nighttime nitrate radical production rate $p(NO_3)$ was $2.6 \pm 2.4$ ppbv $h^{-1}$, and the $\tau(N_2O_5)^{-1}$ was in the range of $0.20\text{-}1.46 \times 10^{-2}$ $s^{-1}$ corresponding to a nighttime

$N_2O_5$ lifetime of 1.1-10.7 min. We also observed a decrease of $\tau(N_2O_5)$ under high relative humidity (RH >40%) conditions due to the higher $N_2O_5$ uptake rates with higher available surface area and liquid water content. $N_2O_5$ and $ClNO_2$ were positively correlated, although the $ClNO_2/N_2O_5$ ratios changed significantly from 0.3 to 95.5. The high $ClNO_2/N_2O_5$ ratios in this study might result from high $ClNO_2$ production rate and fast $N_2O_5$ loss due to the sufficient chloride source supply.

    The $N_2O_5$ uptake coefficients estimated on basis of the product formation rates of $ClNO_2$ and $NO_3^-$ were 0.017-0.09 in

this study. Correspondingly, the direct $N_2O_5$ loss rates via the heterogeneous uptake were in the range of 0.00044-0.0034 $s^{-1}$, contributing 6.9%-32.6% to the total $N_2O_5$ loss. Our results indicated fast $N_2O_5$ loss in the nocturnal boundary in urban Beijing was mainly due to the indirect pathways through $NO_3$ reactions with NO/VOCs rather than the heterogeneous uptake of $N_2O_5$. We also noticed that the derived $ClNO_2$ production yields (0.10-0.35) were substantially lower than those from the Bertram-Thornton parameterization, indicating that future studies are needed to address these discrepancies.

**Acknowledgments**

    This work was supported by the National Natural Science Foundation of China (41571130034, 91744207). The University of Manchester work was supported through the NERC grants for AIRPOLL and AIRPRO (NE/N007123/1, NE/N00695X/1).



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



**Table 1.** Summary of average meteorological parameters (RH, $T$, WS), CIMS species ($N_2O_5$, $ClNO_2$, the calculated $NO_3$, nitrate radical production rate $p(NO_3)$, $N_2O_5$ lifetime ($\tau N_2O_5$) and $NO_3$ lifetime ($\tau NO_3$)), trace gases ($O_3$, $NO_2$, NO), and NR-PM$_1$ species ($NO_3^-$, $Cl^-$) for the entire study and four nighttime periods (i.e., P1, P2, P3 and P4).

| | Entire | P1 | P2 | P3 | P4 |
|---|---|---|---|---|---|
| Meteorological parameters | | | | | |
| RH (%) | 36.8 | 36.3 | 41.3 | 60.5 | 28.0 |
| $T$ ($^o$C) | 26.7 | 24.5 | 23.2 | 23.2 | 29.4 |
| WS (m s$^{-1}$) | 2.9 | 1.9 | 2.3 | 1.9 | 3.7 |
| CIMS species | | | | | |
| $N_2O_5$ (pptv) | 79.2 | 176.2 | 515.8 | 37.8 | 88.3 |
| $ClNO_2$ (pptv) | 174.3 | 427.3 | 748.3 | 227.7 | 57.2 |
| $NO_3$(cal) (pptv) | 8.9 | 7.2 | 48.1 | 2.0 | 18.2 |
| $P(NO_3)$ (ppbv h$^{-1}$) | 3.2 | 3.6 | 2.8 | 1.7 | 2.6 |
| $\tau N_2O_5$ (min) | 1.8 | 1.3 | 8.5 | 1.2 | 1.1 |
| $\tau NO_3$ (min) | 3.3 | 1.8 | 39.8 | 2.5 | 4.3 |
| Gaseous species | | | | | |
| $O_3$ (ppbv) | 51.1 | 23.4 | 55.6 | 17.8 | 40.3 |
| $NO_2$ (ppbv) | 28.1 | 56.2 | 16.9 | 38.2 | 28.7 |
| NO (ppbv) | 8.7 | 15.6 | 0.5 | 2.3 | 7.1 |
| NR-PM$_1$ species | | | | | |
| $NO_3^-$ | 2.7 | 2.3 | 4.3 | 4.3 | 0.6 |
| $Cl^-$ | 0.10 | 0.13 | 0.09 | 0.08 | 0.04 |



**Table 2.** Estimated uptake coefficient of $N_2O_5$, $ClNO_2$ production yield and related parameters for the selected periods at three nights.

| Period | RH (%) | $\gamma N_2O_5$ | ø | $K_d$ (s$^{-1}$) | Percentage (%) |
|--------|--------|-----------------|-----|------------------|----------------|
| Case1 | 39.9 | 0.017 | 0.35 | 0.00044 | 32.6 |
| Case2 | 63.6 | 0.090 | 0.10 | 0.0034 | 20.8 |
| Case3 | 21.1 | 0.019 | 0.15 | 0.00055 | 6.9 |





**Figure 1.** Time series of trace gases ($O_3$, $NO$, $NO_2$), meteorological parameters (RH, $T$, WS, WD), and IAP-CIMS species ($N_2O_5$, $ClNO_2$). The UoM-CIMS and BBCEAS measurements are also shown for inter-comparisons. The four nights (i.e., P1, P2, P3 and P4) are marked for further discussions.





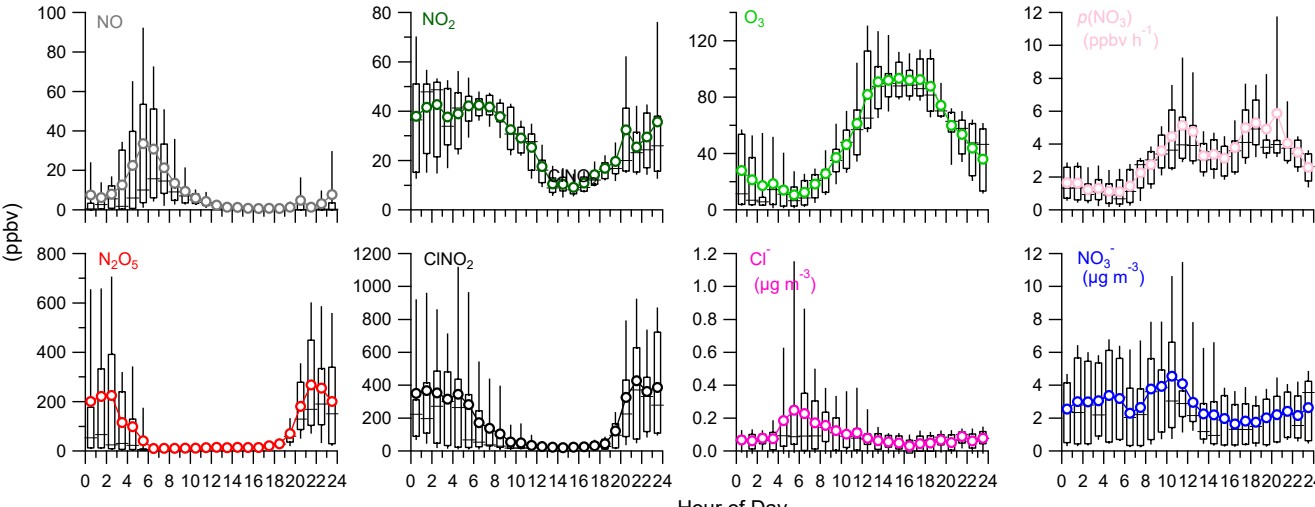

**Figure 2.** Diurnal variations of trace gases (NO, $NO_2$, $O_3$), IAP-CIMS species ($N_2O_5$, $ClNO_2$), nitrate radical production rate $p(NO_3)$, and NR-PM$_1$ species (Cl$^-$, $NO_3^-$).



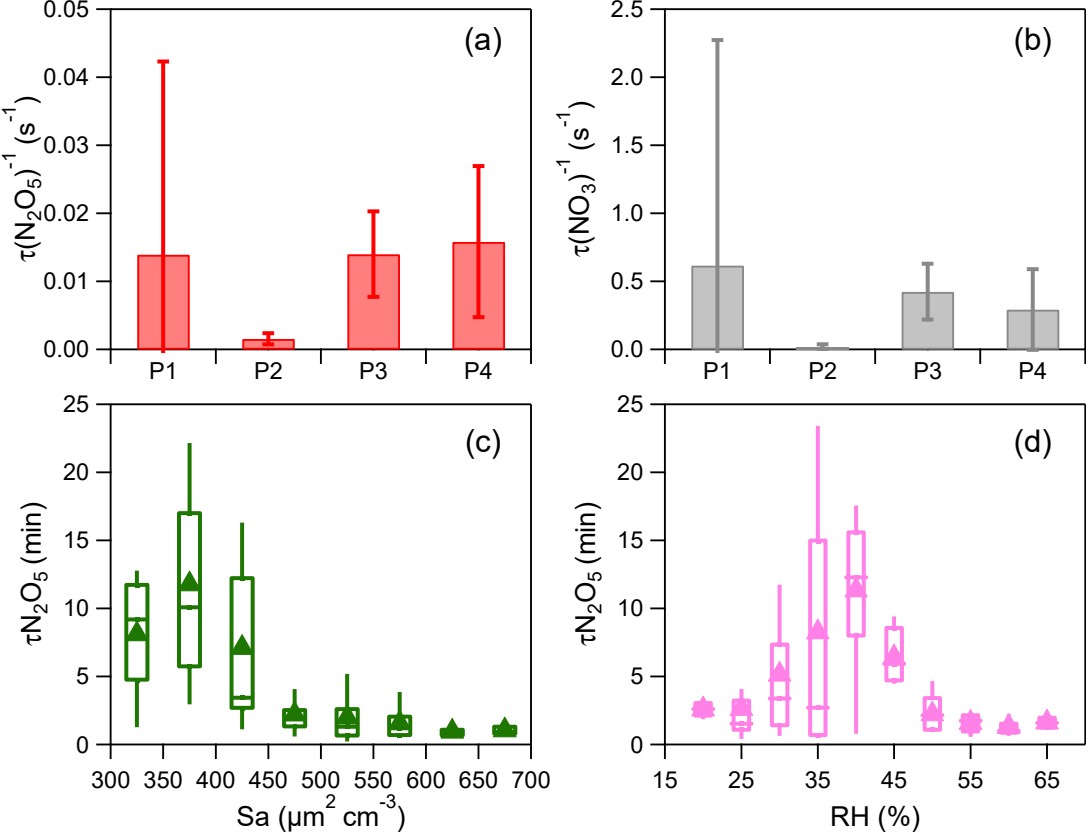

**Figure 3.** (a-b) Average reactivity of $N_2O_5$ ($\tau(N_2O_5)^{-1}$) and $NO_3$ ($\tau(NO_3)^{-1}$) for different nights (i.e., P1, P2, P3 and P4). The error bar represents the standard deviation, (c) Variations of the nocturnal $\tau(N_2O_5)$ as a function of aerosol surface area density ($S_a$), (d) Variations of the nocturnal $\tau(N_2O_5)$ as a function of relative humidity (RH). The data were binned according to $S_a$ (50 $\mu m^2$ $cm^{-3}$ increment) or RH (5% increment). Mean (triangle), median (horizontal line), $25^{th}$ and $75^{th}$ percentiles (lower and upper box), and $10^{th}$ and $90^{th}$ percentiles (lower and upper whiskers) are shown for each bin.





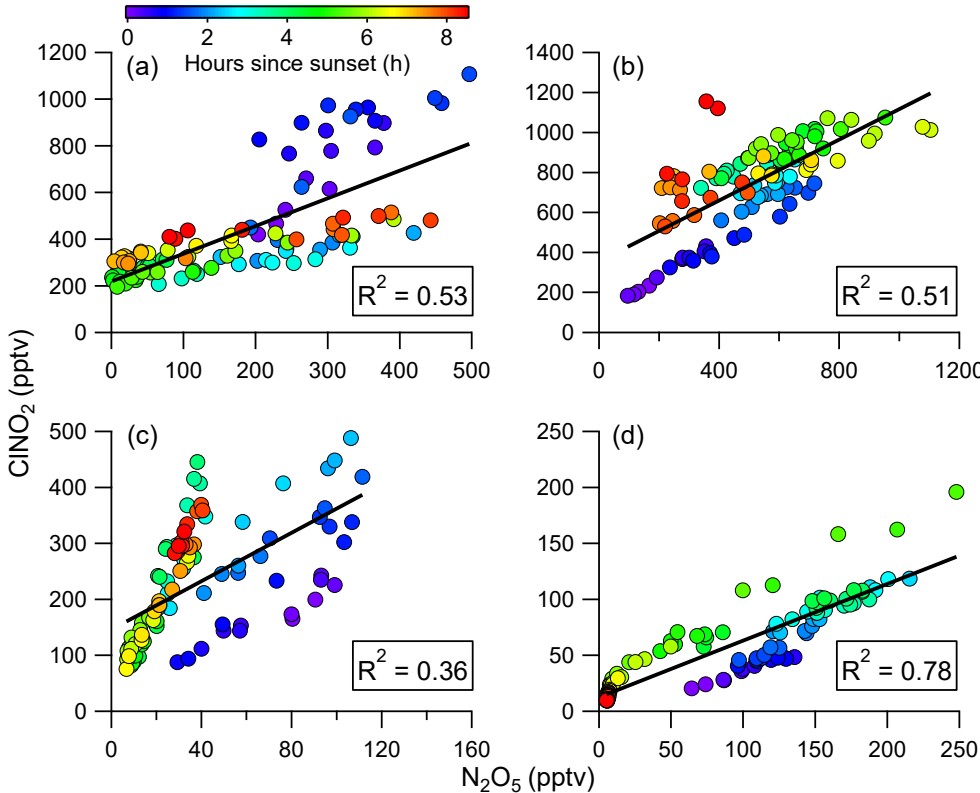

**Figure 4.** Correlations between $ClNO_2$ and $N_2O_5$ for four different nights, i.e., P1, P2, P3 and P4. The data are color-coded by the hours since sunset.



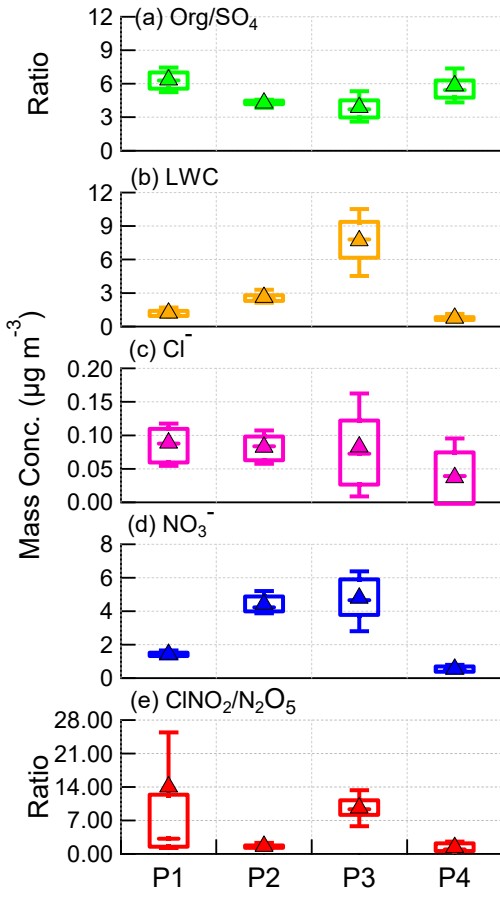

**Figure 5.** Box plots of (a) Org/SO$_4$ ratio, (b) LWC, (c) particulate chloride, (d) particulate nitrate, and (e) ClNO$_2$/N$_2$O$_5$ ratio

for each night, i.e., P1, P2, P3 and P4. The mean (triangle), median (horizontal line), 25[th] and 75[th] percentiles (lower and

upper box), and 10[th] and 90[th] percentiles (lower and upper whiskers) are shown.





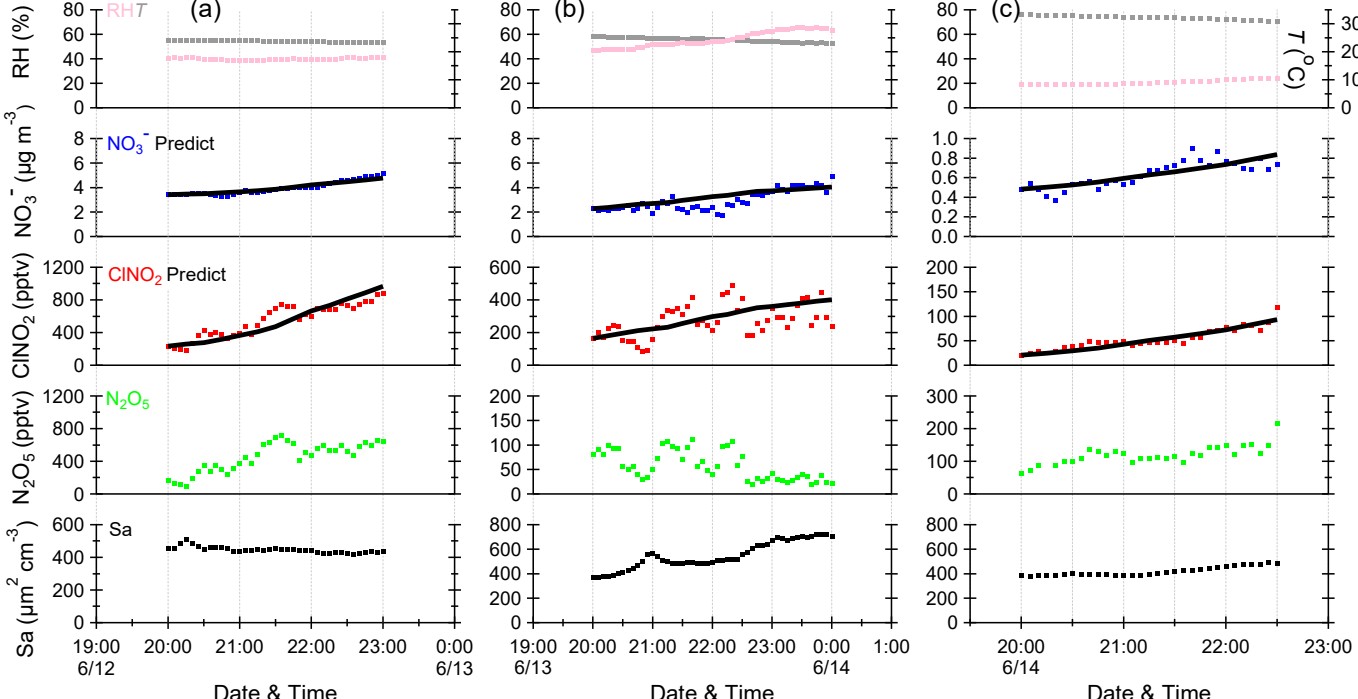

**Figure 6.** Time series of meteorological parameters (RH, $T$), particulate nitrate ($NO_3^-$), mixing ratios of $N_2O_5$ and $ClNO_2$, and aerosol surface area density ($S_a$) for the selected periods at three nights. The black solid lines are the predicted, integration concentrations of $NO_3^-$ and $ClNO_2$ calculated using the estimated method.