# Peer review of "Production of N2O5 and ClNO2 in summer in urban Beijing, China"

_Atmospheric Chemistry and Physics, 2018_

## Referee Comment (RC1) · Anonymous Referee #1 · 27 Apr 2018

Review of "Production of N2O5 and ClNO2 in summer in urban Beijing, China" by Zhou et al. (acp-2018-349)

This manuscript presents measurements of ambient N2O5 and ClNO2 in urban Beijing using chemical ionization mass spectrometry and derivertization of the uptake coefficient of N2O5 and the yield of ClNO2. The data set are certainly of interest to the atmospheric chemistry community. On the other hand, major issues like instrument calibration, size of the data set, and presentation of the results, etc. stopped this reviewer from recommending publication of this manuscript in its present form in Atmospheric Chemistry and Physics. The authors are suggested to address the following concerns before a further consideration can be given.

Main issues

1. The authors are suggested to be consistent in the presentation of their results. Take the abstract for example, $\tau(N2O5)$-1 has been used whereas $\tau(N2O5)$ is given in Table 1; The exact values for $\tau(N2O5)$-1 in the abstract is different from the values in the main text (Page 15 Line 15); Scientific notation has been used with $\tau(N2O5)$-1 but not with direct N2O5 loss rates (0.00044-0.0034 s-1); Finally, the contribution of heterogeneous uptake of N2O5 (7-33%) cannot be derived from the above-mentioned numbers. These certainly hurts the readability of this manuscript.

2. (Page 5 Line 20), I don't agree with the expression that BBCEAS was deployed for inter-comparison of N2O5. The IAP-CIMS was not calibrated at all. To me, BBCEAS provided a calibration reference for the IAP-CIMS. Also, as stated by the authors, BBCEAS measures the sum of N2O5 and NO3. How did they determine NO3 and subtract the values of NO3 subsequently? Please elaborate.

3. (Page 6 Line 1-2), the campaign is quite short, which could be still fine, but the authors are suggested to be more conservative with their findings. (Page 15 Line 13-15), expand the discussion in the time needed for the steady state assumption, and justify whether this requirement was met in the current study. (Page 17 Line 23-24), explain and justify why these three particular time periods are selected.

4. (Page 6 Line 12-21), what were total ion counts of the reagent ions for the IAP-CIMS? Given the high affinity of I- with multiple species in the urban air, was reagent ion depletion observed during the campaign? Was the zero point regularly measured with the IAP-CIMS during the campaign? What were the detection limits and sensitivity of the IAP-CIMS for this particular method? While sensitivity of IAP-CIMS might be derived from comparison with other instruments, how to determine the detection limits? How would this affect the lower points in the measurements?

5. (Page 8-9), a lot of description was given for the calibration of UoM-CIMS but the key is that the IAP-CIMS was not. I still think that it might be OK with the current reference method. But, do consider the uncertainty caused by the assumptions during the entire

process. I would like to see that the authors add a new session to evaluate the potential impact on their general conclusions (say, the relative importance of different pathways) due to this uncertainty (e.g., 10% or 20% uncertainties in the calibration factors).

6. (Page 9 Line 13), elaborate "this calibration was scaled to those in the field. . ."

7. (Page 11 Line 7-9), do the authors mean that ambient particles were dried and then measured with the SMPS? Where did the hygroscopic growth factor come from?

8. (Page 11 Line 16), why is $\tau(N2O5)$-1 defined as the ration of p(NO3), instead of P(N2O5), to the N2O5 mixing ratio?

9. (Page 13 Line 13-14), If this is true, why didn't we see high ClNO2?

10. (Figure 2), if the steady state assumption was met, are we able to derive conc. Of NO3 at least for two hours per day?

11. (Page 15 Line 7), how was Cl2 measured? Was Cl2 calibrated?

12. (Figure 4), I would like to see Figure S6 instead of Figure 4 here. The data points are quite scattered and hence the attempt to use a single linear regression for all the data points just does not make sense.

13. Check the references thoroughly. For example, Brown et al. 2003a in the main text whereas Brown, S.S., . . .2013a in the reference list.

14. (Table 1), add the range or standard deviation in addition to the average values.

15. (Table 2 and the corresponding main text), there are limited number of data points so that statistically we can't draw any conclusion for sure, e.g., the effects of RH (page 18 Line 16-17).

Minor issues

16. (Page 2 Line 2), "on the following day"?

17. (Page 4 Line 14-17), do we really want to name this methodology as I-CIMS?

Personally I prefer iodine adduct CIMS. Also, the authors are suggested to put more effective numbers with m/z values since it is ToF-CIMS after all (Page 7 Line 10-15). Finally, do we really know where the electron/charge is attached? (Page 7 Line 10-15)

18. (Page 6 Line 13), drawn "into" the sampling room?

19. (Page 7 Line 3), so it is CH3I in N2?

20. (Page 12 Line 4), do we want to add "nighttime formation"?

21. (Page 12 Line 21), are those reported numbers averages of 1-min average, or 5-min average, or 30-min average?

22. (Page 13, Line 21-), units for quite many numbers are missing.

23. (Page 14 Line 16-18), a good correlation between NO and black carbon does not necessarily mean NO is the most scavenger for N2O5.

24. (Page 19 Line 2), also include indirect N2O5 loss via titration by NO.

25. (Figure 3c & 3d), repeat the figure caption "the data were binned according . . ." in the main text to help the readers understand how these two plots are derived.

---

## Referee Comment (RC2) · Anonymous Referee #2 · 8 Jun 2018

The authors report four nights of N2O5 and ClNO2 observations in summer at an urban site of Beijing, China. The data were analyzed to show the concentration levels and N2O5 reactivity, and the N2O5 uptake coefficient and ClNO2 product yield were estimated from the field data. This manuscript provides a new piece of measurement data as well as some insights into the nocturnal N2O5 chemistry in the polluted atmosphere of North China. However, the current paper lacks some important details about the measurement and calculation methods, and some interpretation of the measurement results needs to be refined. Overall, this manuscript can be considered for publication after the following specific comments being addressed.

Major Comments:

Further details are required to clarify the quality assurance and quality control of the

[Figure]

N2O5 and ClNO2 measurements.

-The two CIMS systems were not in-situ calibrated during the measurement campaign. The UoM-CIMS was calibrated by the synthesized N2O5 and ClNO2 after the campaign, and the IAP-CIMS was not calibrated and only inter-compared with the BBCEAS instrument. The sensitivity of the CIMS instruments may vary with the different operation conditions. Could the authors comment on the uncertainty of the post-campaign calibration on the present N2O5 and ClNO2 observations.

-The inlet chemistry, including the potential loss of N2O5 and formation of ClNO2 in the sampling inlet, is an important issue in the field measurements of N2O5 and ClNO2, especially for the highly polluted areas such as the study site in the present study. Have the authors checked the inlet issue during the present measurements.

-The background of the CIMS instrument was determined by passing dry N2 to the system in this study. The authors should provide a figure to show the background determination results, maybe in the supplementary materials. In addition, the authors may also need consider to check the instrument zero by adding excess NO to the ambient air, because the dry N2 may be different from the real ambient conditions.

-It has been proposed that the ambient RH may affect the sensitivity of the CIMS to the target compounds. This may affect the analysis results of dependence of N2O5 reactivity on RH. The authors are suggested to further check the potential influence of ambient RH on their CIMS measurements.

-In view of the above issues, the authors should provide an overall estimation of their N2O5 and ClNO2 measurements, at least including the detection limits and uncertainties.

On the calculation and analysis of the N2O5 reactivity:

-It seems that there were no VOC measurements in this study. It is not clear how the authors calculated k(NO3) and then N2O5 reactivity without the VOC data? If the

VOC measurements were available, the authors should provide the concentrations and chemical compositions of major VOC species.

-NO plays a very important role in the nocturnal N2O5 chemistry. Only a considerable level of NO (e.g., >1 ppbv) can significantly suppress the NO3 and then N2O5, as the reaction of NO3 with NO is very fast. This is why the concentrations of N2O5 and ClNO2 are usually low at surface sites in urban areas such as the study site in the present study. In comparison, the oxidation reactions of NO3 and VOCs are relatively slow, and NO3 can only oxidize a small group of specific VOCs, mainly biogenic VOCs and some oxygenated VOCs. The authors argued that the reactions of NO3 with VOCs are important for the N2O5 reactivity. It is better if the authors could separately evaluate the NO3 reactivity towards NO and VOCs.

-The authors assumed a steady-state for NO3 and N2O5 and estimated the lifetimes for these compounds (see Table 1). It is very strange that the lifetime of N2O5 was much shorter than that of NO3 radical. In general, the lifetime of NO3 radical is quite short, but N2O5 may have relatively longer lifetimes during the nighttime.

-Page 12, Lines 1-3: the Equation (6) was only valid if the observed nitrate increase was thoroughly contributed by the in-situ chemical production and the heterogeneous uptake of N2O5 contributed to 100% of the nighttime nitrate formation. The authors need consider the impacts of regional transport and other nitrate formation pathways on this calculation. As mentioned by the authors, previous studies suggested that the heterogeneous uptake of N2O5 only accounted for about 50-100% of nighttime nitrate formation. The authors at least should mention the assumption and limitation of this calculation method.

Page 13, Lines 20-22: this argument is not really true. The N2O5 production potential in P1 should be low because of its very high NOx levels. It is also a little bit strange that the concentrations of N2O5 and ClNO2 are moderately high given such high levels of NOx (>15 ppbv) in P1, but it is a very interesting result. What is the possible reason for

this?

Page 13 Line 25 to Page 14 Line 2: this interpretation is not correct. The difference in the observed N2O5 and ClNO2 concentrations between P2 and P4 should be due to the difference in the NO levels, i.e., 0.5 versus 7.1 ppbv. Given your estimated short lifetimes of NO3 and N2O5, meteorological conditions and transport should not be the major factors here.

Page 15, Lines 6-11: on the low particulate chloride and its weak correlation with ClNO2, another possible reason is the size distribution of chloride aerosol. Only the chloride in PM1 was measured in this study, and it may largely underestimate for the total particulate chloride. Could the authors check the size distribution of chloride from the previous measurements available in urban Beijing and discuss its impacts on the observed results in this study.

Page 15, Lines 14-16 and 20-22: it was not clear how the N2O5 and NO3 reactivities were calculated without the VOC data. It would be better if the authors could also calculate the reactivity from heterogeneous N2O5 uptake, NO3+NO and NO3+VOCs, and compare them among each other.

Page 15, Lines 22-24: I guess that the higher N2O5 reactivity in P4 than P2 should be due to the higher NO level. The authors are encouraged to examine the detailed budget of N2O5 reactivity for both cases and find the exact reason for this.

Page 16, Lines 1-2: it is interesting that the N2O5 reactivity presents a non-linear dependence on aerosol surface area and RH. What are the possible reasons for this?

Page 16, Lines 6-14: it is interesting (and also strange) for the sharp decrease in the N2O5 reactivity with ambient RH from 40% to 50%. As mentioned above, the authors are suggested to examine the dependence of the CIMS sensitivity on the ambient RH.

Page 17, Lines 1-4: the authors are suggested to elaborate more about the air mass transport and its impacts on the observed N2O5 and ClNO2. What is the difference in

the air mass origins among the four cases? Which air masses contained higher N2O5 and ClNO2?

Page 18 Line 24 to Page 19 Line 4: as mentioned above, the reactions of VOCs and NO3 are relatively slow, and NO3 can only react with some specific VOC compounds. In comparison, the titration of NO3 by NO is rather fast. Given the high NO levels observed in urban Beijing in this study, the NO3 loss should be dominated by the NO titration.

Minor Comments:

Page 2, Line 6, "79.2 and 174.3 pptv": pay attention to the use of significant digits. What is the detection limit of the N2O5 and ClNO2 measurements in the present study? Could it be up to 0.1 pptv? Please check and revise the usage of significant digits throughout the manuscript.

Page 2, Lines 6-8: does the N2O5 reactivity here include its indirect loss by NO3? If so, the high N2O5 reactivity may not suggest the large nocturnal nitrate formation potential. Besides the heterogeneous reactions of N2O5, the nitrate formation also depends on the NO3 reactivity and ClNO2 product yield. After all, the authors also pointed out that the N2O5 loss was mainly attributed to the indirect loss by NO3 (Page 2 and Lines 11-13).

Page 3, Line 2 "an efficient sink for the nocturnal removal of nitrogen oxides": "sink" is redundant with "removal", please rephrase this sentence.

Page 3, Lines 2-4: I suggest to separate this long sentence into two short ones, with one defining N2O5 and the other describing its thermal equilibrium with NO3.

Page 3, Lines 4-5: I recall that the reactions of NO3 with VOCs are not very fast. The N2O5 and NO3 removal is mainly attributed to the rapid titration of NO3 by NO in the high NOx environments.

Page 3, Line 9: it should be particulate NO3-, other than HNO3.

[Figure]

Page 3, Line 11: delete "N2O5" as only ClNO2 can be subject to photolysis to release NO3 and chlorine atom.

Page 3, Line 16: ClNO2 product yield. . .

Page 4, Lines 5-7: on the inconsistency between field-derived N2O5 uptake coefficient and the lab-derived parameterizations, the authors should acknowledge the work of Brown et al. 2006.

Brown, S. S., et al.: Variability in nocturnal nitrogen oxide processing and its role in regional air quality, Science, 311, 67-7-, 2006.

Page 4, Lines 10-12: regarding this indirect measurement approach, what technique was used for the measurement of NO3 radical?

Page 4, Lines 13-15 and 17-19: please also refer to the following measurement works of N2O5 and ClNO2 by CIMS in China.

Tham Y. J., et al.: Presence of high nitryl chloride in Asian coastal environment and its impact on atmospheric photochemistry, Chinese Sci. Bull., 59, 356-359, 2014.

Wang T., et al.: Observations of nitryl chloride and modeling its source and effect on ozone in the planetary boundary layer of southern China, J. Geophys. Res., 121, 2457-2475, 2016.

Tham Y. J., et al.: Significant concentrations of nitryl chloride sustained in the morning: investigations of the causes and impacts on ozone production in a polluted region of northern China, Atmos. Chem. Phys., 16, 14959-14977, 2016.

Page 4, Lines 8-19: the description of the commonly used measurement techniques for N2O5 and ClNO2 is incomplete here. The authors need also briefly introduce the Cavity Ring-Down Spectroscopy (CRDS) and the CIMS with an unheated inlet configuration (235 m/z).

Page 4, Line 22: change "several" to "some", as there have been about a dozen measurement studies of N2O5 and ClNO2 in China.

Page 5, 1-2: besides these measurement efforts, recently, some modeling studies have also evaluated the impacts of N2O5 and ClNO2 chemistry on the ozone formation and regional air quality in China. The authors should consider to include these efforts to enrich the current understanding of the nocturnal nitrogen chemistry and its impacts.

Xue L. K., et al.: Development of a chlorine chemistry module for the Master Chemical Mechanism. Geosci. Model Develop. 8. 3151-3162, 2015.

Wang T., et al.: Observations of nitryl chloride and modeling its source and effect on ozone in the planetary boundary layer of southern China, J. Geophys. Res., 121, 2457-2475, 2016.

Li Q. Y., et al.: Impacts of heterogeneous uptake of dinitrogen pentoxide and chlorine activation on ozone and reactive nitrogen partitioning: improvement and application of the WRF-Chem model in southern China, Atmos. Chem. Phys., 16, 14875-14890, 2016.

Page 5, Line 9: delete "However"

Page 5, Lines 15-17: a recent modeling study has evaluated the impacts of heterogeneous ClNO2 formation on the next-day ozone formation in Beijing.

Xue L. K., et al.: Ground-level ozone in four Chinese cities: precursors, regional transport and heterogeneous processes, Atmos. Chem. Phys., 14, 13175- 13188, 2014.

Page 6, Line 7: provide standard deviations for the average values of temperature and RH.

Page 8, Lines 3-5: how did you estimate this uncertainty?

Page 9, Line 20: was the slope of 1.42 derived from the least square regression method? Such slope indicates an average difference of 42% between the two CIMS instruments. Which one gave higher concentrations?

Page 10, Lines19-20: k(N2O5) is commonly used to refer to the heterogeneous reaction rate of N2O5, other than the uptake rate coefficient.

Page 13, Line 2: at Mt. Tai. . .

Page 13, Lines 6-14: please provide the observed concentrations levels of NO and NO2, and also discuss the impact of NOx on the observed variations of N2O5 and ClNO2. As mentioned above, NOx play a very important role in the variability of N2O5 and ClNO2.

Page 13, Line 10: residual boundary layer. . .

Page 13, Lines 15-18: as introduced in the introduction, there have been many studies of N2O5 and ClNO2 in both North China Plain (e.g., Mt. Tai, Beijing, Wangdu, Jinan) and Hong Kong. It would be better if the authors could compare the observed results in this study to these previous results. Is there any difference between the NCP region and Hong Kong in southern China?

Page 13, Lines 20-21: provide the units for 2.8 and 3.6.

Page 14, Line 13: rephrase this sentence. Is there any relationship between the N2O5 formation and the decrease in p(NO3)? p(NO3) is only dependent on the abundances of both O3 and NO2. If anything, the decrease in p(NO3) should weaken the N2O5 formation.

Page 15, Lines 3-4: the reference of Riedel et al. 2012 is not relevant here. It was conducted in US, not in Beijing.

Page 15, Line 6: high emissions from human activities. . .

Page 15, Lines 13-14: it is not clear why only the two-hour data after sunset was used here. Please clarify.

Page 15, Lines 18-19: provide the numbers for the N2O5 loss in southern China and USA for easy comparison.

Table 1: provide the standard deviations and units for the aerosol species.

Figure 1: provide the time series of the aerosol surface area concentrations.

Figure 2: provide the units for N2O5 and ClNO2.

Figure 4: provide the slopes for the regression analysis.

Figure 6: plot the wind sectors to show if the metrological conditions were stable.
* * *

---

## Author Comment (AC1) · 18 Jul 2018

We are thankful to the two reviewers for their thoughtful comments and suggestions. We have revised the manuscript accordingly. Listed below are our point-by-point responses in blue to each reviewer's comments.

**Response to Reviewer #1**

Comments:

This manuscript presents measurements of ambient $N_2O_5$ and $ClNO_2$ in urban Beijing using chemical ionization mass spectrometry and derivertization of the uptake coefficient of $N_2O_5$ and the yield of $ClNO_2$. The data set are certainly of interest to the atmospheric chemistry community. On the other hand, major issues like instrument calibration, size of the data set, and presentation of the results, etc. stopped this reviewer from recommending publication of this manuscript in its present form in Atmospheric Chemistry and Physics. The authors are suggested to address the following concerns before a further consideration can be given.

Main issues:

1.    The authors are suggested to be consistent in the presentation of their results. Take the abstract for example, $\tau(N_2O_5)^{-1}$ has been used whereas $\tau(N_2O_5)$ is given in Table 1; The exact values for $\tau(N_2O_5)^{-1}$ in the abstract is different from the values in the main text (Page 15 Line 15); Scientific notation has been used with $\tau(N_2O_5)^{-1}$ but not with direct $N_2O_5$ loss rates (0.00044-0.0034 $s^{-1}$); Finally, the contribution of heterogeneous uptake of $N_2O_5$ (7-33%) cannot be derived from the above-mentioned numbers. These certainly hurts the readability of this manuscript.

Thank the reviewer's carefulness. We checked the results in the revised manuscript. $\tau(N_2O_5)^{-1}$ represents the reactivity of $N_2O_5$, while $\tau(N_2O_5)$ is the steady-state lifetime. Following the reviewer's suggestions, we changed the two rows of $\tau(N_2O_5)$ and $\tau(NO_3)$ in Table 1 to $\tau(N_2O_5)^{-1}$ and $\tau(NO_3)^{-1}$ to avoid confusion and inconsistency.

$\tau(N_2O_5)^{-1}$ mentioned in the main text (Sec. 3.2), i.e., from $0.16\times10^{-2}$ $s^{-1}$ to $1.58\times10^{-2}$ $s^{-1}$ is the average value for each night, while that in the abstract (from $0.20\times10^{-2}$ to $1.46\times10^{-2}$ $s^{-1}$) refers to the instantaneous values throughout the campaign. We revise the sentence in the corresponding main text to make it clear. Now it reads:

"High $N_2O_5$ reactivity was observed and the average $\tau(N_2O_5)^{-1}$ was 0.16-1.58 $\times10^{-2}$ s$^{-1}$ during these four nights corresponding to a short nighttime $N_2O_5$ lifetime between 1.1 and 10.7 minutes (Fig. 3), with $\tau(N_2O_5)^{-1}$ ranging from $0.20\times10^{-2}$ to $1.46\times10^{-2}$ s$^{-1}$ throughout the campaign. "

The scientific notation, for example, $\tau(N_2O_5)^{-1}$ is generally used in the reference. Comparatively, the direct $N_2O_5$ loss cannot be ubiquitously expressed as a uniform scientific notion, for example, $k_{N_2O_5}$ or $k_d$. That is one of the reasons that $k_{N_2O_5}$ or $k_d$ was not used in the abstract when no detailed information was given in the context.

The contribution of heterogeneous uptake of $N_2O_5$ (7-33%) was calculated according to the $\tau(N_2O_5)^{-1}$ and direct $N_2O_5$ loss ($k_{N_2O_5}$):

$$\tau(N_2O_5)^{-1} = \frac{p(NO_3)}{[N_2O_5]} \approx \frac{k_{NO_3}}{K_{eq}[NO_2]} + k_{N_2O_5}$$

Where $\frac{k_{NO_3}}{K_{eq}[NO_2]}$ denotes the contribution to $\tau(N_2O_5)^{-1}$ from the indirect $N_2O_5$ loss, while $k_{N_2O_5}$ indicates the direct loss of $N_2O_5$ through heterogeneous uptake. The contribution of heterogeneous uptake of $N_2O_5$ is the ratio of $k_{N_2O_5}$ to $\tau(N_2O_5)^{-1}$.

Without the VOCs measurements, the equation above is a robust assessment of the relative contribution between the direct and indirect loss pathway of $N_2O_5$. Furthermore, the uncertainty of this assessment is given in the responses below.

2. (Page 5 Line 20), I don't agree with the expression that BBCEAS was deployed for inter-comparison of $N_2O_5$. The IAP-CIMS was not calibrated at all. To me, BBCEAS provided a calibration reference for the IAP-CIMS. Also, as stated by the authors, BBCEAS measures the sum of $N_2O_5$ and $NO_3$. How did they determine $NO_3$ and subtract the values of $NO_3$ subsequently? Please elaborate.

Thanks for the reviewer's comments. Yes, the $N_2O_5$ sensitivity for IAP-CIMS was derived by comparing with the measurements from BBCEAS, instead of direct calibration. The sum of $N_2O_5+NO_3$ was measured by BBCEAS. We did not subtract the partial of $NO_3$ because the mixing ratio of $N_2O_5$ is much higher than $NO_3$ by a factor of ~11 by applying the equilibrium between $N_2O_5$ and $NO_3$. Due to the lack of equilibrium verification at the daytime, we use the sum of $N_2O_5$ and $NO_3$ for IAP-CIMS

$N_2O_5$ calibration in the study, which leads to an uncertainty of ~17% for $N_2O_5$ associated with the error of $N_2O_5+NO_3$ measurement (~14%). These results suggest that using the sum of $N_2O_5$ and $NO_3$ from BBCEAS for $N_2O_5$ calibration without subtracting $NO_3$ is acceptable.

Furthermore, the estimated $N_2O_5$ for IAP-CIMS tracked well with that measured by UoM-CIMS, and the regression slope (~1.42) was within the uncertainty of $N_2O_5$ measured and calibrated by UoM-CIMS (~58%). So, the inter-comparison between IAP-CIMS and UoM-CIMS further verify the reliability of $N_2O_5$ calibration for IAP-CIMS.

3.    (Page 6 Line 1-2), the campaign is quite short, which could be still fine, but the authors are suggested to be more conservative with their findings. (Page 15 Line 13-15), expand the discussion in the time needed for the steady state assumption, and justify whether this requirement was met in the current study. (Page 17 Line 23-24), explain and justify why these three particular time periods are selected.

Thanks for the reviewer's suggestions. Although the campaign is short, our dataset is statistically reliable as suggested in Fig. R1. Therefore, the conclusions in this study are representative to some extent. We agree that the findings reported here should be careful to expand generally because the field campaign is limited to a short sampling time and influenced by different emission sources.

[Figure]

Figure R1. The box plot of mean (triangle), median (horizontal line), 25th and 75th percentiles (lower

and upper box), and 10th and 90th percentiles (lower and upper whiskers) for $N_2O_5 + NO_3$ during the campaign. Note that "all" refers to the whole dataset measured by BBCEAS from 22 May to 26 June, while "4" refers to the four days overlapped by IAP-CIMS from 12-15 June.

Higher temperature ($> 20\ ^{\circ}C$), $NO_2$ and $O_3$ concentrations ($\sim$ dozens of ppbv) suggest the more rapid steady state time than the simulation results (Brown et al., 2003). Also, the fast $N_2O_5$ and $NO_3$ lifetime in this study is similar to that in Wangdu where only data during the 0.5 h after sunset was used for calculation (Tham et al., 2016). Refer to the time required for steady-state in the literature, for example, 3 h in Hong Kong with much higher $N_2O_5$ lifetime (Brown et al., 2016), the first two hours after sunset were excluded in this study. Although lack of direct steady-state verification, the conclusions in this study are conservative.

The three periods used for calculation were selected for these reasons: (1) concurrent increases in $ClNO_2$ and particulate $NO_3^-$; (2) relatively stable air masses (stable wind direction and relative humidity); (3) no strong fresh emission (e.g., low NO).

4. (Page 6 Line 12-21), what were total ion counts of the reagent ions for the IAP-CIMS? Given the high affinity of I- with multiple species in the urban air, was reagent ion depletion observed during the campaign? Was the zero point regularly measured with the IAP-CIMS during the campaign? What were the detection limits and sensitivity of the IAP-CIMS for this particular method? While sensitivity of IAP-CIMS might be derived from comparison with other instruments, how to determine the detection limits? How would this affect the lower points in the measurements?

Thank the reviewer. The average ($\pm\sigma$) total counts of the reagent ions were about $4\pm0.5\times10^4$ cps (counts per second) ranging from $2.1–5.5\times10^4$ cps during the campaign. Note that the deviation of I- signals was mainly associated with the pressure fluctuation in IMR and SSQ chambers instead of depletion by target molecules, which means that I$^-$ was sufficient during this summer measurements. However, there would be a great possibility that the reagent ion being depleted during polluted periods in the winter of urban Beijing.

We didn't do the zero point measurement but observed the background. For IAP-CIMS, the gas phase background was determined once during the five-day campaign by overflowing the inlet with dry $N_2$

for 35 minutes and the reported concentrations were derived by subtracting the background level in the instrument or the sampling tubes.

For IAP-CIMS, the $N_2O_5$ sensitivity (0.54 cps/pptv) was derived by comparing with the measurement from BBCEAS, while the $ClNO_2$ sensitivity was assumed to be similar with $N_2O_5$. The estimated $ClNO_2$ for IAP-CIMS agrees well with that was measured and calibrated post campaign by UoM-CIMS, suggesting that our reported $ClNO_2$ concentration for IAP-CIMS is reliable.

The detection limit was determined by the three times standard deviation of background measurement and then applied the estimated sensitivity. Although there are data points lower than the detection limit (1.66 pptv for $N_2O_5$ and 0.73 pptv for $ClNO_2$) in the daytime, the average concentrations of $ClNO_2$ and $N_2O_5$ were not much affected. Moreover, we mainly focus on the four nighttime episodes with much higher concentrations than the detection limits.

5.    (Page 8-9), a lot of description was given for the calibration of UoM-CIMS but the key is that the IAP-CIMS was not. I still think that it might be OK with the current reference method. But, do consider the uncertainty caused by the assumptions during the entire process. I would like to see that the authors add a new session to evaluate the potential impact on their general conclusions (say, the relative importance of different pathways) due to this uncertainty (e.g., 10% or 20% uncertainties in the calibration factors).

Good suggestions. The quantifications of $N_2O_5$ and $ClNO_2$ for IAP-CIMS were determined by cross calibration with the BBCEAS and UoM-CIMS. The uncertainty of UoM-CIMS calibration is 58% determined from two $ClNO_2$ calibration methods, which can be used as the uncertainty of $ClNO_2$ measurement. Refer to the literature, the wet surface area density is estimated to be ~ 30% (Wang et al., 2017b;Wang et al., 2018). The uncertainty of $k_{N_2O_5}$ is calculated to be ~35%, while the uncertainty of $\tau(N_2O_5)^{-1}$ is estimated to be ~18% associated from the error of $O_3$ and $NO_2$ (~5%), and $N_2O_5$ (~17%). So, the uncertainty of the direct $N_2O_5$ loss rates contributions estimated from Eq. (2) is ~ 40%.

$$k_{N_2O_5} = \frac{1}{4} \times c \times S_a \times \gamma_{N_2O_5} \tag{1}$$

$$\tau(N_2O_5)^{-1} = \frac{p(NO_3)}{[N_2O_5]} \approx \frac{k_{NO_3}}{K_{eq}[NO_2]} + k_{N_2O_5} \tag{2}$$

I agree to add the uncertainty discussion in the corresponding main text. Now it reads:

"The direct $N_2O_5$ loss rates estimated from the uptake coefficient were in the range of 0.00044-0.0034 $s^{-1}$, which contributed 7-33% to the total $N_2O_5$ loss with the rest being indirect loss. The uncertainty of the direct $N_2O_5$ loss rates contributions is estimated to be ~40%, associated from $S_a$ (~30%), $O_3$ and $NO_2$ (~5%), and $N_2O_5$ (~17%)."

6. (Page 9 Line 13), elaborate "this calibration was scaled to those in the field…"

The $N_2O_5$ and $ClNO_2$ measured by UoM-CIMS were calibrated post campaign, while only formic acid was calibrated throughout the campaign assuming that the ratio between formic acid and $ClNO_2$ sensitivity remains constant during this period. The $ClNO_2$ and formic acid sensitivities in the laboratory were derived by passing the inlet with known concentrations of these gas mixtures. Then, the field $ClNO_2$ sensitivity was derived by scaling to the formic acid sensitivity carried out in the field and the scaling factor is the relative ratio measured in the laboratory.

7. (Page 11 Line 7-9), do the authors mean that ambient particles were dried and then measured with the SMPS? Where did the hygroscopic growth factor come from?

Aerosol particles were dried by a diffusion silica-gel dryer before sampling into the SMPS. The dried aerosol surface area density was calculated according to the SMPS size distribution, which was then calibrated to the ambient RH condition by using the hygroscopic growth factor suggested by Liu et al. (2013) in Beijing. The RH-related parameterization is as follow:

$$f(RH) = 1 + a \times (\frac{RH}{100})^b \text{ , } a = 8.77, b = 9.74$$

8. (Page 11 Line 16), why is $\tau(N_2O_5)^{-1}$ defined as the ration of $p(NO_3)$, instead of $p(N_2O_5)$, to the $N_2O_5$ mixing ratio?

The source for $NO_3$ is the reaction of $NO_2$ with $O_3$ (R1), and the source for $N_2O_5$ is the further reaction of $NO_3$ with $NO_2$. The latter reaction forms a reversible equilibrium.

$$NO_2 + O_3 \rightarrow NO_3 + O_2, \ k_1 \quad\quad\quad (R1)$$

$$NO_2 + NO_3 + M \leftrightarrow N_2O_5 + M, \ K_{eq} \quad\quad\quad (R2)$$

With the steady-state assumption for $N_2O_5$, the formation and destruction of $N_2O_5$ is equalized.

$$\frac{d[N_2O_5]}{dt} = k_1[NO_2][O_3] - [N_2O_5]\tau(N_2O_5)^{-1} = 0$$

$$\tau(N_2O_5)^{-1} = \frac{k_1[NO_2][O_3]}{N_2O_5} = \frac{p(NO_3)}{N_2O_5}$$

where $\tau(N_2O_5)$ denotes the lifetime of $N_2O_5$, with respect to any sink mechanism, including loss processes for $NO_3$ and $N_2O_5$ (Platt et al., 1984;Brown et al., 2003;Brown et al., 2006).

The only source of $N_2O_5$ is R1, that's why $\tau(N_2O_5)^{-1}$ defined as the ration of $p(NO_3)$ rather than $p(N_2O_5)$, to the $N_2O_5$ mixing ratio.

9.    (Page 13 Line 13-14), If this is true, why didn't we see high $ClNO_2$?

The lowest nighttime average of $N_2O_5$ was observed during P3. Although the $ClNO_2$ concentration was not such high in this study compared with previous studies, owing to relatively low values of $\gamma_{N2O5} \times \emptyset$ (0.006-0.009 vs. 0.008-0.035 in reference) (Mielke et al., 2013;Wang et al., 2018), the higher $ClNO_2$ during P3 than P4 with the reversed $N_2O_5$ concentrations supports that fast heterogeneous hydrolysis of $N_2O_5$ under high RH (~ 60.5%) conditions during P3 could be a reason. Another possible explanation was the lowest precursors during P3, e.g., $NO_2$ and $O_3$, consistent with the lowest $p(NO_3)$ during P3 which indicates low production potential for $N_2O_5$ in terms of radical production rates. We revised the sentence:

"The lowest nighttime average of $N_2O_5$ (~ 38 pptv) was observed during P3 although the $NO_2$ showed much higher concentration than those during P2 and P4, indicating the joint influences of precursors ($NO_2$ and $O_3$). Fast heterogeneous hydrolysis of $N_2O_5$ under high RH (~ 60.5%) conditions during P3 could be another reason, which was supported by the higher $ClNO_2$ during P3 than P4. "

10.   (Figure 2), if the steady state assumption was met, are we able to derive conc. Of $NO_3$ at least for two hours per day?

Yes, the $NO_3$ concentration can be calculated according to the $NO_2$ and $N_2O_5$ concentrations when the steady state assumption between $N_2O_5$ and $NO_3$ was met.

$$NO_2 + NO_3 + M \leftrightarrow N_2O_5 + M, \ K_{eq}$$

$$[NO_3(cal)] = \frac{[N_2O_5]}{K_{eq}[NO_2]}$$

Where $K_{eq}$ is the equilibrium rate constant.

The time series of $NO_3$ was generally similar to $N_2O_5$. In Sec. 3.2, the $NO_3$ reactivity was calculated from the inferred $NO_3$. Note that only the periods two hours later after sunset was selected for calculation, rather than only two hours per day.

11. (Page 15 Line 7), how was $Cl_2$ measured? Was $Cl_2$ calibrated?

$Cl_2$ was detected as $I \cdot Cl_2^-$ at $m/z$ 197, $m/z$ 199 and $m/z$ 201 by CIMS. $Cl_2$ was not calibrated yet, and only the raw signal of $Cl_2$ was used for correlation calculation with $ClNO_2$. The well done mass calibration and high resolution peak fitting allow the accurate measurement of $Cl_2$ raw signals despite the absence of $Cl_2$ calibration.

12. (Figure 4), I would like to see Figure S6 instead of Figure 4 here. The data points are quite scattered and hence the attempt to use a single linear regression for all the data points just does not make sense.

Good suggestions. Although the data points seem to be scattered, the positive linear trend is quite obvious. So, the Figure 4 is interpretable and reasonable. The single linear regression indeed failed to characterize the relationship between $N_2O_5$ and $ClNO_2$, and that's why we divide the nighttime into two periods i.e., before midnight and after midnight, to further explain the correlation differences in different air masses (Figure S6). Besides, considering that the time-dependent relationship between $N_2O_5$ and $ClNO_2$ is more visualized in Figure 4 than Figure S6, the Figure 4 is applied in Sec. 3.3 to illustrate the conclusions.

13. Check the references thoroughly. For example, Brown et al. 2003a in the main text whereas Brown, S.S., … 2013a in the reference list.

Thanks for the review's carefulness. We have checked the references in both the main text and reference list one by one.

14. (Table 1), add the range or standard deviation in addition to the average values.

Good suggestions. We add the standard deviation in Table 1 to represent more variability of the data set.

**Table 1.** Summary of average ($\pm 1\sigma$) meteorological parameters (RH, $T$, WS), CIMS species ($N_2O_5$, $ClNO_2$, the calculated $NO_3$, nitrate radical production rate $p(NO_3)$, $N_2O_5$ reactivity ($\tau(N_2O_5)^{-1}$) and $NO_3$ reactivity ($\tau(NO_3)^{-1}$), trace gases ($O_3$, $NO_2$, NO), and NR-PM$_1$ species ($NO_3^-$, $Cl^-$) for the entire study and four nighttime periods (i.e., P1, P2, P3 and P4).

| | Entire | P1 | P2 | P3 | P4 |
|---|---|---|---|---|---|
| Meteorological parameters | | | | | |
| RH (%) | 36.8±15.9 | 36.3±5.5 | 41.3±2.5 | 60.5±6.5 | 28.0±7.0 |
| $T$ ($^o$C) | 26.7±4.9 | 24.5±1.1 | 23.2±0.7 | 23.2±1.4 | 29.4±2.4 |
| WS (m s$^{-1}$) | 2.9±1.4 | 1.9±0.9 | 2.3±0.7 | 1.9±0.6 | 3.7±1.7 |
| CIMS species | | | | | |
| $N_2O_5$ (pptv) | 79±157 | 176±137 | 516±206 | 38±29 | 88±68 |
| $ClNO_2$ (pptv) | 174±262 | 427±223 | 748±221 | 228±104 | 57±39 |
| $NO_3$(cal) (pptv) | 9±16 | 7±7 | 48±26 | 2±2 | 18±15 |
| $P(NO_3)$ (ppbv h$^{-1}$) | 3.2±2.3 | 3.6±4.2 | 2.8±0.5 | 1.7±1.2 | 2.6±1.4 |
| $\tau(N_2O_5)^{-1}$ (s$^{-1}$) | 0.011±0.017 | 0.014±0.028 | 0.0016±0.0008 | 0.014±0.0063 | 0.016±0.011 |
| $\tau(NO_3)^{-1}$ (s$^{-1}$) | 0.34±0.87 | 0.62±1.66 | 0.021±0.017 | 0.42±0.21 | 0.29±0.30 |
| Gaseous species | | | | | |
| $O_3$ (ppbv) | 51.1±35.4 | 23.4±23.2 | 55.6±5.3 | 17.8±15.3 | 40.3±28.0 |
| $NO_2$ (ppbv) | 28.1±17.1 | 56.2±22.4 | 16.9±3.9 | 38.2±9.9 | 28.7±16.0 |
| NO (ppbv) | 8.7±16.9 | 15.6±14.6 | 0.5±0.7 | 2.3±3.5 | 7.1±13.3 |
| NR-PM$_1$ species | | | | | |
| $NO_3^-$ (μg m$^{-3}$) | 2.7±2.4 | 2.3±1.5 | 4.3±0.7 | 4.3±1.6 | 0.6±0.2 |
| $Cl^-$ (μg m$^{-3}$) | 0.10±0.16 | 0.13±0.14 | 0.09±0.02 | 0.08±0.09 | 0.04±0.07 |

15. (Table 2 and the corresponding main text), there are limited number of data points so that statistically we can't draw any conclusion for sure, e.g., the effects of RH (page 18 Line 16-17).

Thank for the suggestions. There are only three episodes selected for the calculation of $\gamma_{N2O5}$ and ø, which seem to confine the applicability of conclusions. For example, $\gamma_{N2O5}$ appeared to increase from 0.019 to 0.090 with the RH rising from 21.1% to 63.6% from case2 to case3. The $\gamma_{N2O5}$ values were comparable between case1 and case3 at low RH levels (< 40%) although RH differed by a factor of 2. The conclusion was drawn based on the results in this study and further supported by previous findings. We didn't generalize the findings to universal conclusions. Long-term measurements in future for better characterization are needed.

Minor issues:

16.  (Page 2 Line 2), "on the following day"?

Thank the reviewer for pointing this out. We revised this sentence and now it reads:

"…impact on …photochemistry on the following day…"

17.  (Page 4 Line 14-17), do we really want to name this methodology as I-CIMS? Personally I prefer iodine adduct CIMS. Also, the authors are suggested to put more effective numbers with m/z values since it is ToF-CIMS after all (Page 7 Line 10-15). Finally, do we really know where the electron/charge is attached? (Page 7 Line 10-15)

Thank the review's suggestions. The CIMS can use protonated water clusters, acetate, nitrate and iodide as regent ions, of which we called the CIMS using iodide as I-CIMS in this referenced methodology. I think it is interpretable.

Yes, we presented the high-resolution data set for analysis rather than the unit mass resolution since it is ToF-CIMS. The $m/z$ 208 and 210 for $I \bullet ClNO_2^-$, and $m/z$ 235 for $I \bullet N_2O_5^-$ are just nominated $m/z$ values for these species. The effective peak fitting at $m/z$ 208, 210, and 235 are shown in Fig. S1 in the supplement.

The molecules were detected as adduction products with iodide. Although we do not know the charge distribution and chemical structure, it does not have influences on the detection and quantification.

18.  (Page 6 Line 13), drawn "into" the sampling room?

We have revised the sentence following the reviewer's suggestion and changed "drawn inside" to "drawn into".

19. (Page 7 Line 3), so it is $CH_3I$ in $N_2$?

Yes, for the UoM-CIMS it is $CH_3I$ (20 sccm) and $N_2$ (4 slm) gas mixtures produced from the custom-made manifold passing over the Tofwerk x-ray ionization source.

20. (Page 12 Line 4), do we want to add "nighttime formation"?

Thanks for the reviewer's suggestions. Yes, processes at daytime hinder the assumption that $ClNO_2$ and $NO_3^-$ are produced only from the heterogeneous $N_2O_5$ uptake, including the photolysis of $ClNO_2$ and other $NO_3^-$ formation pathways. We add this constraint in the revised manuscript to avoid the confusions. Now it reads:

"Only periods with concurrent nighttime formation of $ClNO_2$ and $NO_3^-$ meet the requirements…"

21. (Page 12 Line 21), are those reported numbers averages of 1-min average, or 5-min average, or 30-min average?

Thanks for pointing this out. The average $N_2O_5$ and $ClNO_2$ mixing ratios were reported in 5-min time resolution in this manuscript if no additional explanations. We still revise this sentence and it reads:

"…with the 5-min average ($\pm 1\sigma$) mixing ratios being…"

22. (Page 13, Line 21-), units for quite many numbers are missing.

Thanks for the reviewer's carefulness. We add the units in the revised manuscript. Now it reads:

"The average nitrate radical production rate $p(NO_3^-)$ was 2.8 and 3.6 ppbv $h^{-1}$ during P1 and P2, respectively, which are both higher than those during P3 and P4 (1.7-2.6 ppbv $h^{-1}$)"

23. (Page 14 Line 16-18), a good correlation between NO and black carbon does not necessarily mean NO is the most scavenger for $N_2O_5$.

Good point. The good correlation between NO and black carbon is presented to illustrate the strong local emission of NO in Beijing. The increasing NO before sunrise concurrent with the decreasing $N_2O_5$ implies the significant indirect $N_2O_5$ loss via titration by NO, however it is not sufficient supporting that NO is the most important scavenger for $N_2O_5$. We revised the sentence by taking away "most" in the manuscript.

24. (Page 19 Line 2), also include indirect $N_2O_5$ loss via titration by NO.

Thanks for the reviewer's suggestions. Yes, the indirect $N_2O_5$ loss pathways also include $NO_3$ titration by NO except for VOCs. Indeed, the reaction of $NO_3$ with NO is much faster than those with VOCs, particularly the high $NO_x$ levels in this study. Except for VOCs, we include indirect $N_2O_5$ loss via titration by NO.

25. (Figure 3c & 3d), repeat the figure caption "the data were binned according…" in the main text to help the readers understand how these two plots are derived.

We thank the reviewer for this suggestion. Although the box plot is a standard stuff, we still add the figure caption in the main text to help understanding the two plots.

"Figure 3c shows the $N_2O_5$ lifetime as a function of surface area density ($S_a$) with the data being binned according to the 50 $\mu m^2$ $cm^{-3}$ $S_a$ increment"

**Response to Reviewer #2**

Comments:

The authors report four nights of $N_2O_5$ and $ClNO_2$ observations in summer at an urban site of Beijing, China. The data were analyzed to show the concentration levels and $N_2O_5$ reactivity, and the $N_2O_5$ uptake coefficient and $ClNO_2$ product yield were estimated from the field data. This manuscript provides a new piece of measurement data as well as some insights into the nocturnal $N_2O_5$ chemistry in the polluted atmosphere of North China. However, the current paper lacks some important details about the measurement and calculation methods, and some interpretation of the measurement results needs to be refined. Overall, this manuscript can be considered for publication after the following specific comments being addressed.

Major Comments:

Further details are required to clarify the quality assurance and quality control of the $N_2O_5$ and $ClNO_2$ measurements.

-The two CIMS systems were not in-situ calibrated during the measurement campaign. The UoM-CIMS was calibrated by the synthesized $N_2O_5$ and $ClNO_2$ after the campaign, and the IAP-CIMS was not calibrated and only inter-compared with the BBCEAS instrument. The sensitivity of the CIMS instruments may vary with the different operation conditions. Could the authors comment on the uncertainty of the post-campaign calibration on the present $N_2O_5$ and $ClNO_2$ observations.

The $N_2O_5$ and $ClNO_2$ measured by UoM-CIMS were calibrated post campaign, while formic acid calibration was running regularly twice daily throughout the campaign. This is relying on the assumption that the ratio of sensitivity between formic acid and $ClNO_2$ remains constant throughout. The twice daily formic acid calibration ensures the stable sensitivity over time. Therefore, the post-campaign of UoM-CIMS $N_2O_5$ and $ClNO_2$ calibration could not introduce significant errors compared to the regularly calibration during the campaign. On the other hand, the operation conditions could be carefully controlled during post calibrations to make sure that they are under similar conditions with ambient measurements (under the same IMR and SSQ pressure, with the same TPS voltages settings, under similar $ClNO_2$ concentrations comparing to the ambient air, and the same

reagent ions levels …). Our estimated $ClNO_2$ for the IAP-CIMS agrees well with that of UoM-CIMS (Slope = 0.903), which suggest the uncertainty of $ClNO_2$ for the IAP-CIMS is within 10%.

We use the sum of $N_2O_5$ and $NO_3$ measured by the BBCEAS for IAP-CIMS $N_2O_5$ calibration in the study. The mixing ratio of $N_2O_5$ is much higher than $NO_3$ by a factor of ~11 by applying the equilibrium between $N_2O_5$ and $NO_3$, which leads to the uncertainty of ~ 10% without subtraction of $NO_3$ concentration. The uncertainty of $N_2O_5$ of IAP-CIMS is estimated to be ~ 17% associated with the error of $N_2O_5+NO_3$ measurement (~ 14%). In addition, the transmission efficiency of $N_2O_5$ for IAP-CIMS also introduce additional uncertainty of $N_2O_5$. Given the regression slope of 1.42 between the IAP-CIMS and UoM-CIMS, the uncertainty of $N_2O_5$ could be up to ~ 42%. Considering the uncertainty between different instruments, the uncertainty of $N_2O_5$ is conservatively estimated to be ~ 17%. Overall, the three independent measurements correlated well with each other, which means that the uncertainty of sensitivies of iodide CIMS systems caused by post-calibration was not a concern for quantifications.

-The inlet chemistry, including the potential loss of $N_2O_5$ and formation of $ClNO_2$ in the sampling inlet, is an important issue in the field measurements of $N_2O_5$ and $ClNO_2$, especially for the highly polluted areas such as the study site in the present study. Have the authors checked the inlet issue during the present measurements.

We didn't check the potential loss of $N_2O_5$ and formation of $ClNO_2$ during this campaign. In fact, we removed the Teflon filter in front of the sampling line after 10 June, which had some influence of the sampled concentrations. After that, we replaced sampling lines with brand new ones, and the inlet issue should be minor considering the very fast residence time of less than 0.4 s within the sampling line, which could be further verified by the inter-comparisons results. We agree with the reviewer that the inlet issue should be evaluated in the future studies.

-The background of the CIMS instrument was determined by passing dry $N_2$ to the system in this study. The authors should provide a figure to show the background determination results, maybe in the supplementary materials. In addition, the authors may also need consider to check the instrument zero by adding excess NO to the ambient air, because the dry $N_2$ may be different from the real ambient

conditions.

Thanks for the reviewer's suggestions. We provide the mass spectra and time series of raw signals of $N_2O_5$ and $ClNO_2$ during the background measurement in the supplement. In addition, we did not check the instrument zero during this measurement, which we should have done and will do in our following studies to make sure better data quality.

[Figure]

**Figure R2.** Mass spectra (unit mass resolution) and time series of raw signals of $N_2O_5$ and $ClNO_2$ during the background measurement.

-It has been proposed that the ambient RH may affect the sensitivity of the CIMS to the target compounds. This may affect the analysis results of dependence of $N_2O_5$ reactivity on RH. The authors are suggested to further check the potential influence of ambient RH on their CIMS measurements.

The ionization efficiency and thus sensitivity of the CIMS is dependent on the RH of the sample. The UoM-CIMS during the measurement period is independent of ambient RH changes through the tuning of the ion optics and introduction of $H_2O$ into the ionization mix so that the threshold required for sensitivity independent of changes in water vapor (Bannan et al., 2015). The well correlations of $N_2O_5$

and ClNO$_2$ between the IAP-CIMS and UoM-CIMS ensure the data quality of the measurement.

Also, we plot the relationship between N$_2$O$_5$ and ClNO$_2$ and the data are color-coded by the ambient RH. Although the different slopes along with the hours after sunset can be explained by the air masses from different regions in the main text, the RH-dependent sensitively might also be a reason.

[Figure]

**Figure R3.** Correlations between ClNO$_2$ and N$_2$O$_5$ for four different nights, i.e., P1, P2, P3 and P4. The data are color-coded by the ambient relative humidity (RH). Also shown are the correlation coefficients and slopes.

-In view of the above issues, the authors should provide an overall estimation of their N$_2$O$_5$ and ClNO$_2$ measurements, at least including the detection limits and uncertainties.

Good suggestions. We provide the uncertainties and detection limits for N$_2$O$_5$ and ClNO$_2$ measurements above. Briefly, the uncertainty is 17% and 58%, detection limit is 1.7 pptv for N$_2$O$_5$ and 0.7 pptv for ClNO$_2$.

On the calculation and analysis of the $N_2O_5$ reactivity:

-It seems that there were no VOC measurements in this study. It is not clear how the authors calculated $k(NO_3)$ and then $N_2O_5$ reactivity without the VOC data? If the VOC measurements were available, the authors should provide the concentrations and chemical compositions of major VOC species.

Thank the reviewer's comments. We do not have the VOCs measurements in this study. The $N_2O_5$ reactivity is defined as the inverse $N_2O_5$ steady state lifetime, which is the ratio of $p(NO_3)$ to the $N_2O_5$ mixing ratio. So, we don't need the VOCs data to calculate $k(NO_3)$ and thus the $N_2O_5$ reactivity, but the data of $O_3$, $NO_2$ and $N_2O_5$.

$$\tau(N_2O_5)^{-1} = \frac{p(NO_3)}{[N_2O_5]} \approx \frac{k_{NO_3}}{K_{eq}[NO_2]} + k_{N_2O_5}$$

-NO plays a very important role in the nocturnal $N_2O_5$ chemistry. Only a considerable level of NO (e.g., >1 ppbv) can significantly suppress the $NO_3$ and then $N_2O_5$, as the reaction of $NO_3$ with NO is very fast. This is why the concentrations of $N_2O_5$ and $ClNO_2$ are usually low at surface sites in urban areas such as the study site in the present study. In comparison, the oxidation reactions of $NO_3$ and VOCs are relatively slow, and $NO_3$ can only oxidize a small group of specific VOCs, mainly biogenic VOCs and some oxygenated VOCs. The authors argued that the reactions of $NO_3$ with VOCs are important for the $N_2O_5$ reactivity. It is better if the authors could separately evaluate the $NO_3$ reactivity towards NO and VOCs.

It is really a good point to evaluate the reactions of $NO_3$ with NO and VOCs separately. However, lacking the VOCs data and direct $NO_3$ measurement limited us from further discussions about this topic so far. As the reviewer mentioned, the reaction of $NO_3$ with NO is much faster than that with VOCs. One previous study in urban Jinan reported that the contribution of $N_2O_5$ loss by VOCs could only be larger than that by NO when NO is negligible, e.g., 16.3% vs. 7.1% (Wang et al., 2017a). The $N_2O_5$ reactivity due to the indirect $NO_3$ loss pathway is mainly attributed to the reaction of $NO_3$ with NO rather than VOCs in the $NO_x$-rich air mass (for example, urban Beijing). We revise the relevant sentences on main cause of $N_2O_5$ reactivity in the manuscript.

-The authors assumed a steady-state for $NO_3$ and $N_2O_5$ and estimated the lifetimes for these compounds

(see Table 1). It is very strange that the lifetime of $N_2O_5$ was much shorter than that of $NO_3$ radical. In general, the lifetime of $NO_3$ radical is quite short, but $N_2O_5$ may have relatively longer lifetimes during the nighttime.

Yes, we agree with the reviewer that the lifetime of $NO_3$ radical is generally shorter than that of $N_2O_5$, as Fig. 3 depicts in the manuscript. In fact, the lifetimes of $NO_3$ radical and $N_2O_5$ in Table 1 were reversed. We revised the values and also added the standard deviations in Table 1.

-Page 12, Lines 1-3: the Equation (6) was only valid if the observed nitrate increase was thoroughly contributed by the in-situ chemical production and the heterogeneous uptake of $N_2O_5$ contributed to 100% of the nighttime nitrate formation. The authors need consider the impacts of regional transport and other nitrate formation pathways on this calculation. As mentioned by the authors, previous studies suggested that the heterogeneous uptake of $N_2O_5$ only accounted for about 50-100% of nighttime nitrate formation. The authors at least should mention the assumption and limitation of this calculation method.

Thanks for the reviewer's suggestions. Previous studies suggested that the heterogeneous uptake of $N_2O_5$ accounted for about 50-100% of nighttime nitrate formation, which is the average results. We selected the periods when the heterogeneous uptake of $N_2O_5$ contributed to 100% of the nighttime nitrate formation for calculation. Also, we expanded the assumption and limitation of this calculation method in this paragraph. Now it reads:

"The production rate of particulate nitrate ($pNO_3^-$) was obtained from HR-AMS measurements assuming that the measured $pNO_3^-$ was totally from production of nitrate by reaction R4 (Phillips et al., 2016). Note that the formation of particulate nitrate from regional transport or via the net uptake of $HNO_3$ to aerosol is not taken into consideration."

Page 13, Lines 20-22: this argument is not really true. The $N_2O_5$ production potential in P1should be low because of its very high $NO_x$ levels. It is also a little bit strange that the concentrations of $N_2O_5$ and $ClNO_2$ are moderately high given such high levels of $NO_x$ (>15 ppbv) in P1, but it is a very interesting result. What is the possible reason for this?

The p($NO_3$) during P1 was the highest among the four nighttime which might indicate the high production potential of $N_2O_5$. However, the $N_2O_5$ concentrations during P1 were lower than those during P2 due to the titration of NO. The much higher $N_2O_5$ concentration during P1 than those during P3 and P4 despite the high NO levels during P1 suggests that higher $O_3$ and $NO_2$ might compensate for the loss by NO.

Page 13 Line 25 to Page 14 Line 2: this interpretation is not correct. The difference in the observed $N_2O_5$ and $ClNO_2$ concentrations between P2 and P4 should be due to the difference in the NO levels, i.e., 0.5 versus 7.1 ppbv. Given your estimated short lifetimes of $NO_3$ and $N_2O_5$, meteorological conditions and transport should not be the major factors here.

Thanks for the reviewer's comments. Yes, the meteorological conditions and regional transport should not play a significant role in $N_2O_5$ and $ClNO_2$ concentrations between P2 and P4 in such a short time. We revised the sentence in the manuscript and now it reads:

"We also note that the $p$($NO_3$) was comparable between P4 and P2 (2.6 pptv vs. 2.8 pptv), yet the $N_2O_5$ and $ClNO_2$ mixing ratios during P4 were much lower, likely due to the difference in NO levels, i.e., 0.5 vs. 7.1 ppbv. The favorable dispersing meteorological conditions with higher wind speed and lower relative humidity in P4 than those in P2 might also be an explanation (Table 1)."

Page 15, Lines 6-11: on the low particulate chloride and its weak correlation with $ClNO_2$, another possible reason is the size distribution of chloride aerosol. Only the chloride in $PM_1$ was measured in this study, and it may largely underestimate for the total particulate chloride. Could the authors check the size distribution of chloride from the previous measurements available in urban Beijing and discuss its impacts on the observed results in this study.

Good suggestions. We plot the average size distribution of particulate chloride during this summer campaign (from HR-ToF-AMS measurements) covering the CIMS measurement periods. As Fig. R4 depicts, the chloride peaked at accumulation-mode (~ 500 nm), while the mass-dependent size distribution above 1000 nm accounts for a small portion. Besides, the undetected fraction of chloride (including refractory and particles larger than 1000 nm) by AMS (e.g., NaCl) is mainly from dust or sea salt particles, which had minor influences on the particulate chloride concentrations in urban Beijing.

[Figure]

Figure R4. Average size distribution of the particulate chloride during the summer campaign from 17 May to 29 June, 2017.

Page 15, Lines 14-16 and 20-22: it was not clear how the $N_2O_5$ and $NO_3$ reactivities were calculated without the VOC data. It would be better if the authors could also calculate the reactivity from heterogeneous $N_2O_5$ uptake, $NO_3+NO$ and $NO_3+VOCs$, and compare them among each other.

The $N_2O_5$ reactivity is defined as the inverse $N_2O_5$ steady state lifetime, which is the ratio of $p(NO_3)$ to the $N_2O_5$ mixing ratio. Similarly, the $NO_3$ reactivity is defined as the ratio of $p(NO_3)$ to the $NO_3$ mixing ratio. Due to the lack of VOCs data, the reactivity from heterogeneous $N_2O_5$ uptake, $NO_3$ with NO and $NO_3$ with VOCs could not be calculated.

$$\tau(N_2O_5)^{-1} = \frac{p(NO_3)}{[N_2O_5]} \approx \frac{k_{NO_3}}{K_{eq}[NO_2]} + k_{N_2O_5}$$

$$\tau(NO_3)^{-1} = \frac{p(NO_3)}{[NO_3]}$$

Page 15, Lines 22-24: I guess that the higher $N_2O_5$ reactivity in P4 than P2 should be due to the higher NO level. The authors are encouraged to examine the detailed budget of $N_2O_5$ reactivity for both cases and find the exact reason for this.

Thanks for the reviewer's suggestions. The higher $N_2O_5$ reactivity in P4 than P2 was due to the higher NO level. We revised the reason in this sentence:

"Note that P2 and P4 showed comparable $p(NO_3)$ (2.8 vs. 2.6 ppbv h$^{-1}$) (Table 1), yet the $N_2O_5$

reactivity during P4 ($1.58\times10^{-2}$ s$^{-1}$) was significantly higher than that during P2 ($0.16\times10^{-2}$ s$^{-1}$) likely due to the higher NO level, and the enhanced $N_2O_5$ heterogeneous loss might also be explanation."

Page 16, Lines 1-2: it is interesting that the $N_2O_5$ reactivity presents a non-linear dependence on aerosol surface area and RH. What are the possible reasons for this?

The $N_2O_5$ lifetime showed an increase as a function of RH at RH< 40%. The other factors, for example, aerosol loading and composition could also have an influence on the $N_2O_5$ uptake (Morgan et al., 2015), thus the $N_2O_5$ lifetime. The exact reason is not clear yet, which should be explored in future studies.

Page 16, Lines 6-14: it is interesting (and also strange) for the sharp decrease in the $N_2O_5$ reactivity with ambient RH from 40% to 50%. As mentioned above, the authors are suggested to examine the dependence of the CIMS sensitivity on the ambient RH.

Thanks for the reviewer's suggestions. The $N_2O_5$ lifetime $\tau(N_2O_5)$ decrease at high RH levels (RH >40%) might be caused by the increased $N_2O_5$ uptake rates due to the higher surface area density ($S_a$). In addition, the increasing aerosol liquid water content at high RH might be another reason. Also, we examine the dependence of the CIMS sensitivity on the ambient RH (see our response above).

Page 17, Lines 1-4: the authors are suggested to elaborate more about the air mass transport and its impacts on the observed $N_2O_5$ and $ClNO_2$. What is the difference in the air mass origins among the four cases? Which air masses contained higher $N_2O_5$ and $ClNO_2$?

We thank the reviewer. The 48 h back trajectories arriving at the sampling site between 19:00-05:00 were calculated every hour using the Hybrid Single Particle Lagrangian Integrated Trajectory (HYSPLIT, NOAA) model. Air masses from the southeast (e.g. P1) usually contain more gaseous pollutants, which resulted in higher concentrations of both $N_2O_5$ and $ClNO_2$, while air masses from the northwest were relatively clean with low levels of pollutants.

During P1 and P4, the air mass was from the similar regions before and after midnight, i.e., southeast during P1 and northeast during P4. During P2, the air mass was originated from the southeast before

midnight and northwest/west after midnight. During P3, the back trajectories were different during the two periods, i.e., before and after midnight. The differences in regression coefficient among the four nights can be explained by different air masses originating from different regions.

Page 18 Line 24 to Page 19 Line 4: as mentioned above, the reactions of VOCs and $NO_3$ are relatively slow, and $NO_3$ can only react with some specific VOC compounds. In comparison, the titration of $NO_3$ by NO is rather fast. Given the high NO levels observed in urban Beijing in this study, the $NO_3$ loss should be dominated by the NO titration.

Yes, the $NO_3$ loss should be dominated by the NO titration, particularly with much high NO concentration in this study. But for the three cases selected for $\gamma_{N_2O_5}$ and ø calculation, the NO concentrations are negligible and the indirect losses towards NO and VOCs might be different. We revise the sentences as following:

"While the uncertainties in different analysis methods, e.g., the product formation rates or steady-state assumption are one of the reasons, the high NO concentration could be the important reason for the dominant $N_2O_5$ loss pathway. The high VOCs emissions, particularly biogenic emissions in summer than other seasons might be another reason for the differences in dominant $N_2O_5$ loss pathway. Indeed, the indirect $N_2O_5$ loss via $NO_3$+VOCs was also found to dominate the total loss of $N_2O_5$ (67%) in summer in suburban Beijing (Wang et al., 2018)."

Minor Comments:

Page2, Line6, "79.2 and174.3pptv": pay attention to the use of significant digits. What is the detection limit of the $N_2O_5$ and $ClNO_2$ measurements in the present study? Could it be up to 0.1 pptv? Please check and revise the usage of significant digits throughout the manuscript.

Thanks much for the suggestions. The mixing ratio of $N_2O_5$ and $ClNO_2$ can be up to the level of 0.1 pptv due to the limit of detection (LOD). We carefully check and revise the usage of significant digits throughout the manuscript.

Page 2, Lines 6-8: does the $N_2O_5$ reactivity here include its indirect loss by $NO_3$? If so, the high $N_2O_5$ reactivity may not suggest the large nocturnal nitrate formation potential. Besides the heterogeneous

reactions of $N_2O_5$, the nitrate formation also depends on the $NO_3$ reactivity and $ClNO_2$ product yield. After all, the authors also pointed out that the $N_2O_5$ loss was mainly attributed to the indirect loss by $NO_3$ (Page 2 and Lines 11-13).

Yes. The $N_2O_5$ reactivity here includes its indirect loss by $NO_3$ and the heterogeneous uptake. The following analysis indicated that the $N_2O_5$ loss was mainly attributed to the indirect loss by $NO_3$ rather than the heterogeneous uptake. Also, the $ClNO_2$ yields derived in this study were not such significant and the $NO_3$ reactions with VOCs and NO were fast. These results together suggest that the nocturnal nitrate formation could be small. We revised this sentence in the manuscript. Now it reads:

"High reactivity of $N_2O_5$, with $\tau(N_2O_5)^{-1}$ ranging from $0.20\times10^{-2}$ to $1.46\times10^{-2}$ $s^{-1}$, suggests active nocturnal chemistry."

Page 3, Line 2 "an efficient sink for the nocturnal removal of nitrogen oxides": "sink" is redundant with "removal", please rephrase this sentence.

Thanks for the reviewer's suggestions. We rephrased the sentence in the revised manuscript as follows:

"Dinitrogen pentoxide ($N_2O_5$) is an effective nocturnal sink for nitrogen oxides…"

Page 3, Lines 2-4: I suggest to separate this long sentence into two short ones, with one defining $N_2O_5$ and the other describing its thermal equilibrium with $NO_3$.

We agree with the reviewer to separate this into two short ones. Now it reads:

"Dinitrogen pentoxide ($N_2O_5$) is an efficient nocturnal sink for nitrogen oxides ($NO_x$) (Dentener and Crutzen, 1993; Brown et al., 2006). $N_2O_5$ exists in a rapid temperature-dependent thermal equilibrium with nitrate radical ($NO_3$) – one of the most important oxidants at night-time (Wayne et al., 1991)."

Page 3, Lines 4-5: I recall that the reactions of $NO_3$ with VOCs are not very fast. The $N_2O_5$ and $NO_3$ removal is mainly attributed to the rapid titration of $NO_3$ by NO in the high $NO_x$ environments.

Thanks for the reviewer's ideas. The reactions of $NO_3$ with VOCs are slower than that with NO. We just listed the possible loss pathway of $N_2O_5$ and $NO_3$ in this sentence rather than compared the loss

frequency.

Page 3, Line 9: it should be particulate $NO_3^-$, other than $HNO_3$.

The gas-particle partitioning of $HNO_3$ form particulate $NO_3^-$. To avoid the ambiguity, we change "$HNO_3$" to "particulate nitrate" in this the sentence.

Page 3, Line 11: delete "$N_2O_5$" as only $ClNO_2$ can be subject to photolysis to release $NO_3$ and chlorine atom.

Thanks for the reviewer carefulness. We delete "$N_2O_5$" in this sentence in the revised manuscript.

Page 3, Line 16: $ClNO_2$ product yield...

Thank the reviewer. We change the "$ClNO_2$ yield" in this sentence into "$ClNO_2$ product yield".

Page 4, Lines 5-7: on the inconsistency between field-derived $N_2O_5$ uptake coefficient and the lab-derived parameterizations, the authors should acknowledge the work of Brown et al. 2006. Brown, S. S., et al.: Variability in nocturnal nitrogen oxide processing and its role in regional air quality, Science, 311, 67-7-, 2006.

Thank the reviewer. We add the work of Brown et al. 2006 to the reference list.

Page 4, Lines 10-12: regarding this indirect measurement approach, what technique was used for the measurement of $NO_3$ radical?

The $NO_3$ radical was measured in one unheated channel. Thermal conversion of $N_2O_5$ to $NO_3$ in a second, heated channel provides simultaneous measurements of the sum of $NO_3$ and $N_2O_5$. The measurement of $N_2O_5$ is obtained via the difference between the two channels. Also, the collocated measurement of $NO_2$ and temperature can also be used for $NO_3$-$N_2O_5$ equilibrium, if without the unheated channel.

Page 4, Lines 13-15 and 17-19: please also refer to the following measurement works of $N_2O_5$ and $ClNO_2$ by CIMS in China.

Tham Y. J., et al.: Presence of high nitryl chloride in Asian coastal environment and its impact on

atmospheric photochemistry, Chinese Sci. Bull., 59, 356-359, 2014.

Wang T., et al.: Observations of nitryl chloride and modeling its source and effect on ozone in the planetary boundary layer of southern China, J. Geophys. Res., 121, 24572475, 2016.

Tham Y. J., et al.: Significant concentrations of nitryl chloride sustained in the morning: investigations of the causes and impacts on ozone production in a polluted region of northern China, Atmos. Chem. Phys., 16, 14959-14977, 2016.

Thanks for the reviewer's suggestions. We have referred to the measurement works of $N_2O_5$ and $ClNO_2$ by CIMS in China in the following lines in this paragraph. But, we can also add these measurement works in the revised manuscript.

Page 4, Lines 8-19: the description of the commonly used measurement techniques for $N_2O_5$ and $ClNO_2$ is incomplete here. The authors need also briefly introduce the Cavity Ring-Down Spectroscopy (CRDS) and the CIMS with an unheated inlet configuration (235 m/z).

Thank the reviewer. We expanded the descriptions of the commonly used measurement techniques for $N_2O_5$ and $ClNO_2$. Now it reads:

"For example, $N_2O_5$ can be derived from the thermal equilibrium with $NO_2$ and $NO_3$ that are simultaneously measured by differential optical absorption spectroscopy (DOAS) (Platt and Stutz, 2008;Stutz et al., 2004). Another indirect measurement of $N_2O_5$ is subtracting ambient $NO_3$ from the total measured $NO_3$ after converting $N_2O_5$ to $NO_3$ in a heated inlet and then detected by Cavity Ring-Down Spectroscopy (CRDS), Cavity-Enhanced Absorption Spectroscopy (CEAS) or Laser-Induced Fluorescence (LIF) (O'Keefe and Deacon, 1988;Smith et al., 1995;Brown et al., 2001;Wood et al., 2003;Stutz et al., 2010). The simultaneous indirect measurements of $N_2O_5$ and $NO_3$ can be implemented using thermal dissociation – chemical ionization mass spectrometer (TD – CIMS) with high sensitivity and time resolution (Stutz et al., 2004), although the interference of m/z 62 ($NO_3^-$) from thermal decomposition of peroxy acetyl nitrate (PAN) and other related species need to be considered (Wang et al., 2014). Recently, the CIMS using iodide reagent ions (I-CIMS) with an unheated inlet configuration allowed the direct measurements of $N_2O_5$ (Kercher et al., 2009;Tham et al., 2014;Wang et al., 2016;Tham et al., 2016). "

Page 4, Line 22: change "several" to "some", as there have been about a dozen measurement studies of $N_2O_5$ and $ClNO_2$ in China.

We change "several" to "some" in this sentence for more rigorous wording.

Page5, 1-2: besides these measurement efforts, recently, some modeling studies have also evaluated the impacts of $N_2O_5$ and $ClNO_2$ chemistry on the ozone formation and regional air quality in China. The authors should consider to include these efforts to enrich the current understanding of the nocturnal nitrogen chemistry and its impacts.

Xue L. K., et al.: Development of a chlorine chemistry module for the Master Chemical Mechanism. Geosci. Model Develop. 8. 3151-3162, 2015.

Wang T., et al.: Observations of nitryl chloride and modeling its source and effect on ozone in the planetary boundary layer of southern China, J. Geophys. Res., 121, 24572475, 2016.

Li Q. Y., et al.: Impacts of heterogeneous uptake of dinitrogen pentoxide and chlorine activation on ozone and reactive nitrogen partitioning: improvement and application of the WRF-Chem model in southern China, Atmos. Chem. Phys., 16, 14875-14890, 2016.

Thank the reviewer for proving these modeling studies. We add the listed reference after this sentence to enrich the current understanding of the nocturnal nitrogen chemistry and its impacts. Now it reads:

"Besides these measurement efforts, recently, some modeling studies have also evaluated the impacts of $N_2O_5$ and $ClNO_2$ chemistry on the ozone formation and regional air quality in China (Xue et al., 2015;Wang et al., 2016;Li et al., 2016). Despite this…"

Page 5, Line 9: delete "However"

Yes, we delete "However" in the revised sentence.

Page 5, Lines 15-17: a recent modeling study has evaluated the impacts of heterogeneous $ClNO_2$ formation on the next-day ozone formation in Beijing.

Xue L. K., et al.: Ground-level ozone in four Chinese cities: precursors, regional transport and heterogeneous processes, Atmos. Chem. Phys., 14, 13175- 13188, 2014.

Thank the reviewer. We add the listed reference and change the sentence in Lines 15-17. Now it reads:

"A recent modeling study has evaluated the impacts of heterogeneous $ClNO_2$ formation on the next-day ozone formation in Beijing (Xue et al., 2014). However, the role of $N_2O_5$ in nitrate formation and of $N_2O_5$ and $ClNO_2$ in night- and day-time chemistry in summer in urban Beijing during filed campaign are not characterized yet, except for one measurement…"

Page 6, Line 7: provide standard deviations for the average values of temperature and RH.

Thanks for the reviewer's suggestions. We provide the standard deviations for the average values of temperature and RH. Now it reads:

"The hourly average RH ranged from 12.9% to 82.8%, with an average value of 36.8±15.9%, and the hourly average temperature ranged from 17.9$^o$C to 38.7$^o$C, averaged at 26.7±4.9$^o$C."

Page 8, Lines 3-5: how did you estimate this uncertainty?

For the BBCEAS, a poly tetrafluoroethylene (PTFE) filter of pore size 1 μm was used to remove aerosol particles from the air stream. Because of aging effects of particles, the filter is typically change at several hours intervals. Besides, the high $NO_x/NO_y$ ratio (~0.78) suggests that the plumes observed in the campaign were primarily from the local urban area rather than the aged air masses from regional transport. So, the less oxidized particles and regularly changed filter ensure the insignificant influence of particle aging.

Page 9, Line 20: was the slope of 1.42 derived from the least square regression method? Such slope indicates an average difference of 42% between the two CIMS instruments. Which one gave higher concentrations?

Yes, the slope of 1.42 was derived from the linear regression. The $N_2O_5$ of IAP-CIMS showed higher concentrations than that of UoM-CIMS.

Page 10, Lines19-20: k($N_2O_5$) is commonly used to refer to the heterogeneous reaction rate of $N_2O_5$, other than the uptake rate coefficient.

Yes, we revised this sentence as following:

"…where $k_{N_2O_5}$ is the heterogeneous reaction rate of $N_2O_5$, and …"

Page 13, Line 2: at Mt. Tai...

Thank the reviewer. Yes, we change the sentence in the revised manuscript by changing "in Mt. Tai…" to "at Mt. Tai…".

Page 13, Lines 6-14: please provide the observed concentrations levels of NO and $NO_2$, and also discuss the impact of $NO_x$ on the observed variations of $N_2O_5$ and $ClNO_2$. As mentioned above, $NO_x$ play a very important role in the variability of $N_2O_5$ and $ClNO_2$.

Thanks for the reviewer's suggestions. The average levels of NO and $NO_2$ during the four nights are shown in Table 1. We discuss the impact of $NO_x$ on the observed variations of $N_2O_5$ and $ClNO_2$ in the revised manuscript. Now it reads:

"Besides, the maximal $N_2O_5$ occurred during P2 other than the rest nights was likely due to the insignificant titration of NO during P2, e.g., 0.5 vs. 2.3-15.6 ppbv. The lowest nighttime average of $N_2O_5$ (~ 38 pptv) was observed during P3 although the $NO_2$ showed much higher concentration than those during P2 and P4, indicating the joint influences of precursors ($NO_2$ and $O_3$). Fast heterogeneous hydrolysis of $N_2O_5$ under high RH (~ 60.5%) conditions during P3 could be another reason, which was supported by the higher $ClNO_2$ during P3 than P4."

Page 13, Line 10: residual boundary layer...

Thank the reviewer's carefulness. Yes, we change the sentence in the revised manuscript by changing "residential boundary layer…" to "residual boundary layer...".

Page 13, Lines 15-18: as introduced in the introduction, there have been many studies of $N_2O_5$ and $ClNO_2$ in both North China Plain (e.g., Mt. Tai, Beijing, Wangdu, Jinan) and Hong Kong. It would be better if the authors could compare the observed results in this study to these previous results. Is there any difference between the NCP region and Hong Kong in southern China?

Thanks for the reviewer's suggestions. $ClNO_2$ presented the highest value (1.4 ppbv, 5-minute average) on 13 June, yet it is lower than the maximum of 2.1 ppbv (1-minute average) observed at Wangdu (Tham et al., 2016), 2.9 ppbv (1-minute average) in suburban Beijing (Wang et al., 2018), and also the $ClNO_2$ peak of 2.1 ppbv (1-minute average) at Mt. Tai (Wang et al., 2017b). The ubiquitously high $ClNO_2$ in the NCP are consistent with those reported in Hong Kong, e.g., 4.7 ppbv (1-minute average maximum) (Wang et al., 2016) and 2.0 ppbv (1-minute average maximum) (Tham et al., 2014). There are insignificant differences between the NCP and Hong Kong in terms of the maximum $ClNO_2$ concentration, although the polluted air masses were originated from different sources. For example, the pollution in Hong Kong was transported from inland areas of the PRD, while it came from the power plant and industrial plumes of the NCP at Mt. Tai and outflows of urban Beijing in Wangdu.

Page 13, Lines 20-21: provide the units for 2.8 and 3.6.

We provide the units for the values in this sentence. Now it reads:

"The average nitrate radical production rate p(NO3) was 2.8 ppbv $h^{-1}$ and 3.6 ppbv $h^{-1}$ during P1 and P2, respectively…"

Page 14, Line 13: rephrase this sentence. Is there any relationship between the $N_2O_5$ formation and the decrease in $p(NO_3)$? $p(NO_3)$ is only dependent on the abundances of both $O_3$ and $NO_2$. If anything, the decrease in $p(NO_3)$ should weaken the $N_2O_5$ formation.

Thank the reviewer's suggestions. The variation of $p(NO_3)$ is only dependent on the abundances of both $O_3$ and $NO_2$ rather than the $N_2O_5$ formation. The $p(NO_3)$ can be regarded as the production potential for $N_2O_5$ in terms of radical production rates. So the decrease in $p(NO_3)$ weaken the $N_2O_5$ formation, and that's the reason we compare the average $p(NO_3)$ among the four nighttime when discussing the $N_2O_5$ concentrations. We revised this sentence:

"$N_2O_5$ was rapidly formed after sunset."

Page 15, Lines 3-4: the reference of Riedel et al. 2012 is not relevant here. It was conducted in US, not in Beijing.

We cited the reference of Riedel et al. 2012 to explain the possible chloride source contributing to the formation of $ClNO_2$ in this study, rather than explain the proven pathway observed in previous study in Beijing. But we still remove this reference in this sentence to avoid the misunderstanding.

Page 15, Line 6: high emissions from human activities...

Thanks for the carefulness. We revised the sentence in the revised manuscript as following:

"… gas-phase HCl due to the high emissions from human activities."

Page 15, Lines 13-14: it is not clear why only the two-hour data after sunset was used here. Please clarify.

The $NO_3$ concentration can be calculated according to $NO_2$ and $N_2O_5$ when the steady state assumption between $N_2O_5$ and $NO_3$ was met. Only the periods two hours later after sunset was selected for calculation to consider for the maximum steady state time, rather than only two hours per day.

Page 15, Lines 18-19: provide the numbers for the $N_2O_5$ loss in southern China and USA for easy comparison.

Yes, we add the numbers for the $N_2O_5$ loss in southern China and USA for easy comparison. Now it reads:

"In comparison, the $N_2O_5$ loss is much more rapid than that previously reported in southern China (1-5 h) (Brown et al., 2016) and the USA (a few hours) (Wagner et al., 2013)."

Table 1: provide the standard deviations and units for the aerosol species.

Thank the reviewer's suggestions. We provide the standard deviations for the average values in Table 1.

Figure 1: provide the time series of the aerosol surface area concentrations.

Thanks for the reviewer's suggestions. We add the time series of aerosol surface area concentrations in Figure 1 in the revised manuscript.

[Figure]

Figure 1. Time series of (a-b) meteorological parameters (RH, T, WS, WD) and surface area density (Sa), (c) trace gases ($O_3$, NO, $NO_2$), (d-e) IAP-CIMS species ($N_2O_5$, $ClNO_2$). The UoM-CIMS and BBCEAS measurements are also shown for inter-comparisons. The four nights (i.e., P1, P2, P3 and P4) are marked for further discussions.

Figure 2: provide the units for $N_2O_5$ and $ClNO_2$.

Thanks for the suggestions. We have already provided the units of gaseous species on the left axis. But, we also add the units for each species in Fig.2 for more clear understanding.

[Figure]

Figure 2. Diurnal variations of trace gases (NO, NO$_2$, O$_3$), IAP-CIMS species (N$_2$O$_5$, ClNO$_2$), nitrate radical production rate p(NO$_3$), and NR-PM1 species (Cl$^-$, NO$_3^-$).

Figure 4: provide the slopes for the regression analysis.

Yes, we add the slopes for the regression analysis in Fig. 4.

[Figure]

Figure 4. Correlations between ClNO$_2$ and N$_2$O$_5$ for four different nights, i.e., P1, P2, P3 and P4. The data are color-coded by the hours since sunset. Also shown are the correlation coefficients and slopes.

Figure 6: plot the wind sectors to show if the metrological conditions were stable.

Thanks for the ideas. We add the plot of time series of wind direction in the supplement to prove that the meteorological conditions during the selected periods were relatively stable.

[Figure]

Figure R5. The time series of wind direction for the selected periods at three nights.

Reference

[revised manuscript text omitted]